# Can we detect conditioned variation in political speech? two kinds of discussion and types of conversation

**Sabina J. Sloman** *, **Daniel M. Oppenheimer, Simon DeDeo**

Department of Social and Decision Sciences, Carnegie Mellon University, Pittsburgh, Pennsylvania, United States of America

* ssloman@andrew.cmu.edu

## Abstract

Previous work has demonstrated that certain speech patterns vary systematically between sociodemographic groups, so that in some cases the way a person speaks is a valid cue to group membership. Our work addresses whether or not participants use these linguistic cues when assessing a speaker's likely political identity. We use a database of speeches by U.S. Congressional representatives to isolate words that are statistically diagnostic of a speaker's party identity. In a series of four studies, we demonstrate that participants' judgments track variation in word usage between the two parties more often than chance, and that this effect persists even when potentially interfering cues such as the meaning of the word are controlled for. Our results are consistent with a body of literature suggesting that humans' language-related judgments reflect the statistical distributions of our environment.

**Data Availability Statement:** The data and analysis code that support the findings of this research are openly available at github.com/sabjoslo/talking-politics/tree/master/experiments/conditional-variation.

## Introduction

What can you tell about someone who addresses a group of people as "you guys" versus "yinz," or someone who stresses the vowel sound in the word "aunt" or doesn't pronounce the "r" in "car" [1]?

**Socially conditioned variation** refers to systematic and idiosyncratic shifts in the language used by members of a particular group [2]. Speech patterns that exhibit socially conditioned variation can be used to identify members of a group [3–7]. For instance, in Glasgow, how a person pronounces the letter "T" reliably indicates that person's age [7, 8], and in New York the pronunciation of "r" reveals a number of sociodemographic attributes [5, 9]. While language is a vehicle for explicitly-constructed semantic content, structural and systematic variation in language also conveys information about a speaker's environment and past experiences.

But do listeners take advantage of this variation as a source of social information? Can we learn—without being explicitly taught—to associate glottal stops with younger speakers [8], longer words with male speakers [6, 10], and the phrase "yinz" with Pittsburghers [11]? Recovery of statistically regular patterns is an important part of language acquisition [12–14], suggesting that such learning may be possible.

**Funding:** The author(s) received no specific funding for this work.

**Competing interests:** The authors have declared that no competing interests exist.

Previous work has shown that the relative frequency of linguistic signals can indeed be used to discriminate between members of different demographic groups [6, 15, 16]. More generally, people associate certain "linguistic profiles" with members of different communities and cultural backgrounds—although these profiles can reflect misleading stereotypes as well as systematic variation in speech patterns [17]. Our work examines whether people use socially conditioned variation as a cue to a particular form of social identity: political identity. In particular, we investigate whether or not participants respond to the relative frequency of linguistic signals when categorizing speakers as Democrats or Republicans.

Throughout this paper, we distinguish between the *conditioned variation* in usage and *sense* of a word. A word's *sense* is its meaning, often operationalized as its dictionary definition [18–20]. Cognitively, we think of a word's *sense* as contributing to an inference drawn from the concept conveyed by the word. For example, upon overhearing a politician using a word that conveys a money-related concept, such as "financial" or "monetary," a listener could make an inference about whether the politician is more likely to be a Democrat or a Republican on the basis of the degree to which they associate each party with the concept of money.

However, even when the concept conveyed is held constant, the listener can make an even more informed guess on the basis of the specific word the speaker chose: Did the listener overhear "financial" *or* "monetary"? Although the two words convey the same concept and have very similar definitions, according to our data Democrats use the phrase "financial" more frequently, while Republicans use the phrase "monetary" more frequently. In other words, if they overheard the politician say "financial," the listener should infer that the speaker is more likely to be a Democrat, while if they overheard "monetary," the listener should infer that the speaker is more likely to be a Republican. In this case, the listener would be relying on the conditioned variation in usage of each word.

The phrase "conditioned variation" generally does not convey the nature of the conditioning variable. For example, variation in the pronunciation of the final consonants of words is often a function of the phonetic features they precede or follow [2]. Following Samara, Smith, Brown, and Wonnacott (2017), we refer to linguistic variation that can be anticipated on the basis of demographic or social characteristics of the speaker as *socially conditioned variation* [2]. We use the term *politically conditioned variation* to refer to linguistic variation that can be anticipated on the basis of the speaker's political identity.

The driving question of our work is: Without relying on their *sense* of what the two parties stand for, can people use politically conditioned variation to make accurate categorization judgments—in other words, learn to pick up on the political analog of "yinz"?

## Politically conditioned variation

While other demographic features, such as race, gender and age, can be quickly and accurately identified on the basis of appearance [21], people have virtually no ability to leverage a target's appearance to accurately determine their political affiliation [22]. To the extent that people are sensitive to politically conditioned variation, verbal cues like word choice could be more reliable in determining a target's political affiliation.

Knowing whether people can use linguistic cues to infer a person's political identity has significant practical relevance. In light of research showing that political ideology is a predictor of behavior [23] and interest [24], a general sensitivity to politically conditioned variation could imply that politicians and partisans emit clues about important aspects of their behavior, beliefs and identity without even realizing it. In addition, information about a person's political affiliations can affect how they are treated: Balliet, Tybur, Wu, Antonellis, and Van Lange (2018) found that in a social dilemma game, partisans cooperated more with members of their

political in-group [25]. Knowledge of our sensitivity to politically conditioned variation could help us better understand the kinds and validity of cues we use to make implicit judgments about others (e.g. contributors to our first impressions [26]), and the mechanisms of group formation and appeal (e.g. the effectiveness of techniques such as "dog-whistle politics;" see the general discussion).

However, the fact that listeners often do not know the political affiliation of a speaker might also make it nearly impossible for them to acquire associations between partisan identity and conditioned variation in speech in the first place. While political identity does correlate with observable demographic characteristics, such features are noisy cues of partisanship, and observers may attribute variation in speech patterns to a more salient social category. Thus, while people have been shown to be sensitive to other sources of socially conditioned variation [2, 6, 15, 16], it remains an open question whether this sensitivity extends to ideological categories.

## Detecting signals using NLP

With the advent of readily-accessible, large-scale datasets, many researchers have attempted to isolate linguistic variation conditioned on a variety of social identities [27, 28]. Diermeier, Godbout, Yu, and Kaufmann (2012) [29], Jensen, Naidu, Kaplan, and Wilse-Samson (2012) [30] and Gentzkow, Shapiro, and Taddy (2019) [31] also investigate variations in speech patterns between the two major U.S. political parties. Diermeier et al. (2012) build a support vector machine (SVM) classifier of a speaker's political ideology and perform post-hoc feature analysis to identify the words that were especially informative in the SVM's classification decisions [29]. Jensen et al. (2012) measure the partisanship of trigrams (contiguous sequences of three words) as the correlation between the frequency with which a speaker utters a given trigram and the speaker's political identity [30]. Gentzkow et al. (2019) posit a generative model of speaker phrase choice, and derive a measure of phrase-level partisanship from components of the parameterized model [31].

The aforementioned models have provided crucial convergent evidence that there is reliable and detectable politically conditioned variation in language use. However, the question at hand is whether people are capable of picking up on that signal. To test this, we turn to the work of Preoţiuc-Pietro, Xu, and Ungar (2016) [6] who find that human raters are able to correctly detect sociodemographic characteristic of speakers in 70% of cases. Preoţiuc-Pietro et al. (2016) use a log-odds measure (see the following section) to isolate linguistic variation between speakers of various demographic groups (e.g. age and gender) [6]. Their method provides a roadmap to how best extend investigation into the domain of U.S. political ideology.

Foreshadowing our results, we find sensitivity that is considerably less extreme than the findings of Preoţiuc-Pietro et al. (2016) [6], highlighting that the range of factors in the correspondence of our judgments with statistical variation in word usage is far from completely understood.

## Method

We used the congressional-record project [32] to access the transcripts of all proceedings in the U.S. House of Representatives between 2012 and 2017, made publicly available as part of the U.S. Congressional Record. The Congressional Record was also the basis of the results reported in the three studies on conditioned variation in political speech summarized above [29–31]. One advantage of the Congressional Record is that it is systematically formatted, allowing us to more easily label the text by matching speakers with entries in databases of members' party affiliations. Another advantage the Congressional Record has over other

corpora of political speech, such as transcripts of the U.S. presidential debates, is that it is substantially larger and has a roughly equal balance of Republican and Democratic speech.

Our corpus consisted of data from each of the most recent five calendars years. (We began data collection in 2018, meaning that 2017 was the last complete calendar year before our studies were run). A time window that was too short would not have yielded enough data for us to recover reliable statistical indicators of linguistic divergence. On the other hand, a time window that was too large would have obscured recent divergences, as politically conditioned speech patterns drift over long periods of time [30, 31]. We chose a five-year window on the basis of our judgment that this time frame would provide a reasonable sample of data which had the power to detect and isolate contemporary patterns of politically conditioned variation.

We first assembled all words spoken by a member of Congress identified as a Democrat or a Republican. While "Republican" and "Democrat" do not exhaust the set of possible political identities, they overwhelmingly dominate the party affiliations of congressional representatives. (At the time of writing, the U.S. House of Congress is composed of 232 Democrats, 196 Republicans and 1 Independent [33]). We then coerced all text in this initial corpus to lower-case, and excluded words we determined were unlikely to reflect meaningful or generalizable variation in word usage, e.g. common prepositions and the names of other sitting members of Congress. The full list of exclusion criteria is included in S1 Appendix. The corpus we used for analysis contains 13,523,319 instances of 16,218 unique words (6,924,484 instances spoken by Democrats and 6,598,835 instances spoken by Republicans).

Language can be analyzed and perceived at many different scales of analysis, e.g. phonemes, word forms, phrases and sentences [16, 34, 35]. We conducted our analyses at the word level primarily for feasibility: Word boundaries are much easier to detect by natural language processing algorithms than the boundaries of phonemes or phrases. It is also interesting to note that some argue that the word is the appropriate unit of analysis in linguistic change and language learning [16, 34, 36].

**Measures of relative frequency.** Adapting the approach of Preoţiuc-Pietro et al. (2016) [6], we calculate exact measures of the relative frequency with which a word was spoken by a Democrat [Republican].

We use the log odds that the word was spoken by a member of a given party, shown in Eq 1, as our measure of the conditioned variation exhibited by a word. The conditional probability terms are calculated directly as the empirical probability that a word $w$ was spoken by a Republican [Democrat] according to our corpus.

$$
\begin{aligned}
logodds_R(w) \quad &= log\frac{P(w|R)}{P(w|D)} \\
&= -log\frac{P(w|D)}{P(w|R)} \\
&= -logodds_D(w)
\end{aligned}
\tag{1}
$$

For example, if our corpus of Republican speech contained only the words "quick," "brown" and "brown," and the corpus of Democratic speech contained only the words "brown," "fox" and "fox," $P("brown"|R) = \frac{2}{3}$ and $logodds_R("brown") = log\left(\frac{2/3}{1/3}\right) = log(2) = 1$. To avoid the discontinuities that arise when some probabilities are 0, we incorporated $L1$ smoothing in our measurements, i.e. imputed one "phantom" observation of each word in both the Republican and Democratic distributions.

8,345 of the 16,218 unique words in our corpus had a corresponding $logodds_R > 0$. We refer to these as Republican words. The remaining 7,873 words had a $logodds_R < 0$. We refer to

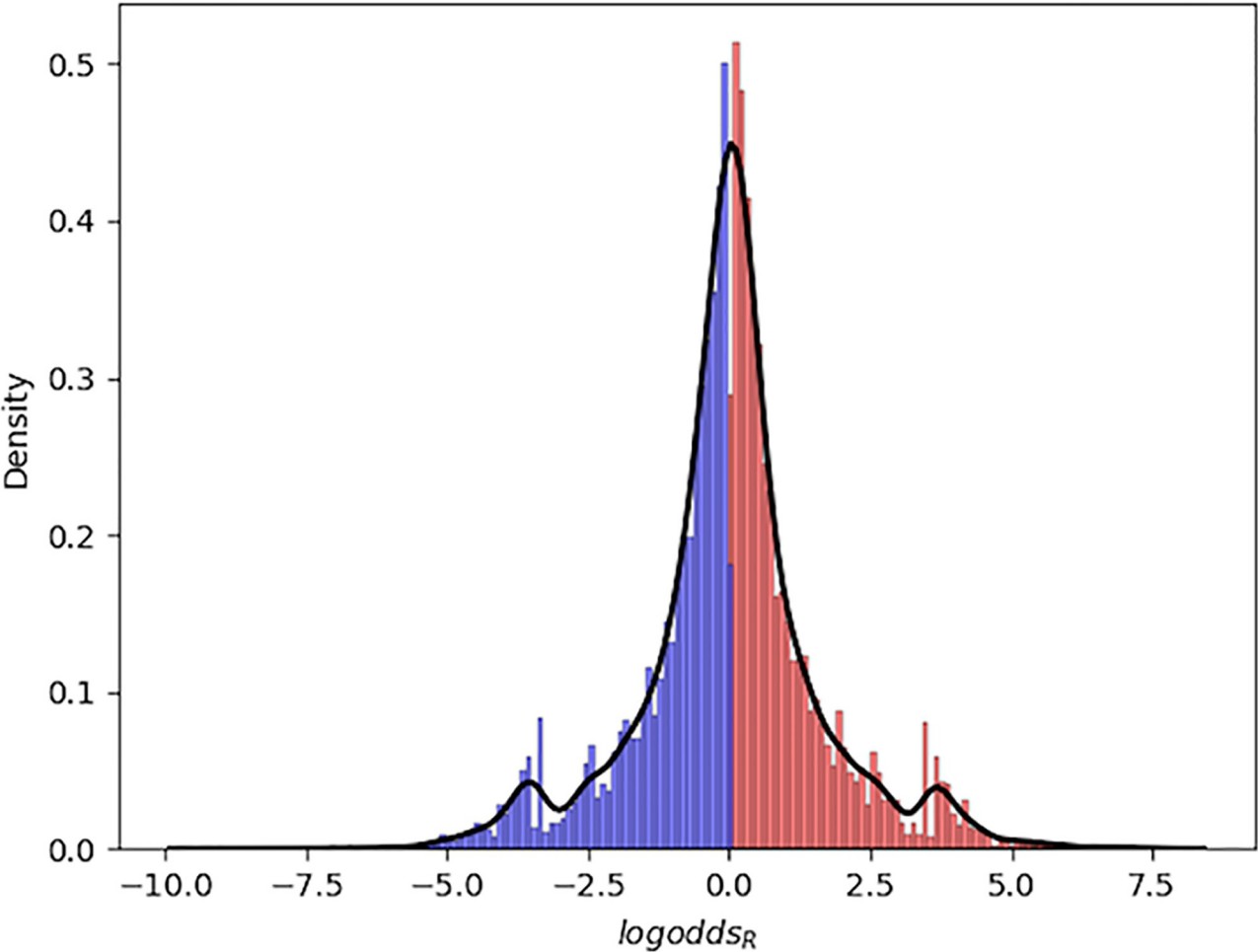

**Fig 1. Distribution of *logodds$_R$*.** Distribution of the values of *logodds$_R$*, our measure of how much more likely a word is to be said by a Republican than by a Democrat, corresponding to each word in our corpus (see text for details; *logodds$_D$* = −*logodds$_R$*). Republican words (*logodds$_R$* > 0) are red, while Democratic words (*logodds$_R$* < 0) are blue. The black line shows the approximate density of the distribution.

these words as Democratic words. The mean *logodds$_R$* value was .02 (*SE* = .01). This was significantly greater than 0 ($t_{16217}$ = 1.96; *p* = .05), indicating that the distribution of *logodds$_R$* was shifted to the right of zero: In the absence of other information, odds were slightly higher that a word was spoken by a Republican. Fig 1 shows the distribution of *logodds$_R$* values.

The *logodds$_R$* measure also closely resembles an element of a traditional model of behavioral response to perceptual inputs. In signal detection theory, the optimal detection threshold is the likelihood ratio of signal to noise: the probability of the stimulus conditional on a signal being present divided by the probability of the stimulus conditional on no signal being present [37, 38].

**Validating *logodds$_R$* as a measure of politically conditioned variation.** Implicit in our operationalization of politically conditioned variation is the assumption that the *logodds$_R$* measure calculated from speech recorded in the Congressional Record captures differentiating patterns in political speech more generally. While we assume that most of our participants have been exposed to political speech by members of both parties, we do not assume that they are regularly exposed to speech on the floor of the U.S. House of Representatives. In this section, we show the extent to which the direction of the political signal—whether or not the word is

more often spoken by a Republican or a Democrat—estimated from the Congressional Record cross-validates to a more public-facing corpus of political speech: the U.S. presidential debates.

We accessed transcripts of all the debates held as part of the 2012 and 2016 presidential election cycles (general and primary, presidential and vice presidential, and main and undercard) from the American Presidency Project [39], and pre-processed them in the same way we pre-processed the raw text from the Congressional Record.

In total, 2,408 words (14.85% of the vocabulary from the Congressional Record corpus) appeared in both the Congressional Record and presidential debates corpora. The correlation between the $logodds_R$ values calculated from the Congressional Record and from the debates is.33 ($t_{2406}$ = 17.412; $p$ <.01). 1,421 of these words (59.01%; $SE$ = 1.00%) have the same estimated polarity (the direction of the sign of the associated $logodds_R$ value) in the two corpora. An exact one-sided binomial test shows that this is significantly greater than chance ($p$ <.01). Overall, there is systematic variation in the speech of Republicans and Democrats that is present in a variety of contexts.

While the correlation between the $logodds_R$ values in the two corpora is highly significant, it is admittedly moderate. While we cannot completely explain the sources of divergence between the distributions in the two corpora, S1 Fig shows that the more politically conditioned variation a word exhibits, the more likely that measure of politically conditioned variation is to generalize across the two corpora. In other words, the stronger the political signal, the more likely it is to operate in both contexts.

## Study 1: Testing alignment of judgments with the direction of politically conditioned variation

Before we can test whether human judgments align with politically conditioned variation when word *sense* is held constant, we first have to determine whether people's judgments align with politically conditioned variation at all. In Study 1, we test this basic intuition by presenting participants with words that were statistically more likely to have been said by a Democrat or a Republican, and ask them to make judgments about the most likely party identity of the speakers of those words.

All studies reported in the following sections were approved by the Carnegie Mellon University Institutional Review Board under IRB IDs STUDY2018_00000167 and STUDY2017_00000367. We obtained electronic consent from all participants.

### Participants

201 subjects completed Study 1 on MTurk. Our use of MTurk as a recruiting tool was driven by two primary considerations: i) convenience and ii) access to a more representative population than we would achieve with in-person samples (even with a local non-university sample, well below 10% of our population would be Republican given the demographics in the city in which we conducted our research). It is worth noting that scholars have documented disadvantages to using MTurk as a recruitment tool, including the possibility of non-naïvety and low quality responses (see Chandler, Mueller, and Paolacci (2014) [40] for a discussion of these issues), although our use of attention and quality checks should have mitigated that to a large degree (see details below). Moreover, a number of scholars have shown that MTurk can yield reliable data [41, 42] (our studies were completed before the "MTurk Crisis" [43] began affecting data quality). However, we cannot rule out the possibility that non-diligent responders corrupted our sample. To the extent that this *is* the case, we believe that would only serve to reduce our power by introducing noise, making our results conservative estimates of the population effect.

After excluding 54 participants for failing the attention check (described in the following section), our analyzed sample contains 147 participants, including 61 self-identifying Democrats and 38 self-identifying Republicans. These exclusions do not affect our main results. In this and all subsequent studies, we restricted participant eligibility to those of voting age residing in the U.S. Participants in the analyzed sample had a mean self-reported age of 37.66 (*SE* = .84), and included 74 men and 72 women (1 participant did not report their gender identity). 81.63% of participants reported having voted in the 2016 presidential election.

## Methods

After completing a demographics questionnaire, participants were presented with a list of words and asked to ". . .estimate how likely it is that the word is spoken either by a Democrat or by a Republican [Republican or by a Democrat]" (the full instructions are included in S2 Appendix).

The words "Democrat" and "Republican" were presented in a random order. Participants rated each word on a 6-point scale, from "I am almost certain the speaker is a Democrat" (which we coded as 1) to "I am almost certain the speaker is a Republican" (which we coded as 6). Each page of the survey contained 20 items. For approximately half of participants the presentation order of response options was reversed.

For Study 1, we wanted to use stimuli that both exhibited significant conditioned variation (had a *logodds*$_R$ with a large magnitude), and were spoken frequently enough by the associated party that participants were likely to have been exposed to the variation in usage. For each word *w*, we calculated the *partial Kullback-Leibler divergence* (*PKL*), a measure that combines the *logodds*$_R$ with the word's probability of occurrence (interested readers can consult Klingenstein, Hitchcock, and DeDeo (2014) [44] for further details). Words with a high *PKL*$_D$ are both more likely to be spoken by Democrats *and* are spoken frequently by Democrats, while words with a high *PKL*$_R$ are both more likely to be spoken by Republicans *and* are spoken frequently by Republicans. In other words, *PKL* isolates strong and frequent statistical signals of party identity.

As stimuli for Study 1, we selected the 39 words with the highest *PKL*$_D$ and the 39 words with the highest *PKL*$_R$. (Elements of the pre-processing pipeline have changed slightly since these stimuli were selected. All analyses were run using the versions of the metrics calculated as described in the previous section). Three words were excluded from analysis of Study 1 since in subsequent pre-processing they were considered to be Congressional-specific stopwords according to the criteria identified in S1 Appendix: *affordable*, *trump* and *obama*. The reported analyses include 37 Democratic words and 38 Republican words.

Two of these words were randomly reselected. If in response to either of these two attention-check stimuli a participant gave a rating that differed more than one point from their original rating on that stimulus, we removed them from our analysis. In total, each participant was presented with 78 stimuli chosen on the basis of the partisanship of the stimulus (39 Democratic words and 39 Republican words) and two words included as attention checks. The full list of stimuli used for all studies is included in S2 Appendix.

All analyses reported in this paper were conducted in the programming languages Python or R [45–48]. We relied on several packages for statistical analyses and visualization, including but not limited to SciPy [49], *scikit-learn* [50] and Plotly [51]. All participant data and code for all of our analyses can be found at https://github.com/sabjoslo/talking-politics.

## Results

The mean judgment on the Republican stimuli was 3.81 (*SE* = .08), just above the indifference point of the scale of 3.5 (participants could express maximum indifference with

responses of "I am unsure but think the speaker is a Republican" or "I am unsure but think the speaker is a Democrat," which we coded as a 4 and a 3, respectively). A one-sided, one-sample $t$-test led us to reject the null that this value was less than or equal to the indifference point ($t_{5580} = 3.77$; $p <.01$). Standard errors reported and used by the inferential tests in this subsection are clustered at the participant and item level using the method in Arai (2011) [52]. Unless stated otherwise, this is the case for results reported in the main results sections for all studies.

The mean judgment on the Democratic stimuli was 3.30 ($SE = .12$), just below the indifference point. A one-sided, one-sample $t$-test led us to reject the null that this value was greater than or equal to the indifference point ($t_{5435} = -1.66$; $p = .05$). A one-sided two-sample $t$-test of the difference in means also led us to reject the null hypothesis that the mean judgment of the Republican words was equal to the mean judgment of the Democratic words ($t_{11015} = 3.50$; $p <.01$).

The results of the inferential tests reported thus far demonstrate that in this experiment, judgments do align with the direction of politically conditioned variation. To understand the relative strength of the effect, we calculated a standardized effect size [53]. Because our data included multiple judgments from each participant, a measure like Cohen's $d$ would be difficult to interpret. We instead used the method of Rouder, Morey, Speckman, and Province (2012) [54] and demonstrated by Westfall (2016) [55]: We used a mixed modeling framework to calculate an effect size, and standardized this effect size by the standard deviation of the residuals of this model. More specifically, we estimated a linear mixed model of the ratings data: We considered the set of each of the individual ratings to be the endogenous variable. We included a fixed effect on a dummy variable indicating whether the word was Democratic or Republican, and considered the estimated coefficient on this variable as our effect size. In addition, we included random effects corresponding to individual participants and items. (We incorporated both participant-level random intercepts, participant-level random slopes on the fixed effect and item-level random intercepts. Since items do not overlap between the groups indicated by our main independent variable—i.e. the sets of Democratic and Republican words are mutually exclusive—it would have been meaningless to specify item-level differences in the fixed effect). We estimated a standardized effect size of.41, which can be interpreted as the effect of the direction of politically conditioned variation on a listener's judgment of the speaker's political identity. (Using a version of the model that includes just the fixed effect— which is equivalent to estimating the traditional Cohen's $d$—we calculated a standardized effect size of.36). We therefore consider the effect of the direction of politically conditioned variation to be of moderate strength [53, 56].

**Item-level analyses.** While clustering standard errors at the participant- and item-level provides some assurance that our effects are not simply driven by high performance on a single item or an especially discriminating participant, it does not completely rule out these possibilities. In the following two subsections, we describe and analyze sensitivity to the direction of politically conditioned variation at the item- and participant-level, respectively. In both sections, we consider a judgment to be accurate if it is above [below] 3.5 and the item is more likely to be said by a Republican [Democrat]. Item-level accuracies are computed on the basis of the mean of all judgments collapsed across participants.

50 of the 75 words (66.67%) were accurately classified (24, or 64.86%, of the Democratic words, and 26, or 68.42%, of the Republican words). The average item-level accuracy was 58.62% ($SE = 2.33$%), which was significantly different from chance performance of 50% ($t_{74} = 3.70$; $p <.01$).

We also ran a Mann-Whitney $U$ test, a non-parametric alternative to the $t$-test. We ranked all 75 words by their associated mean judgment, and tested against the null hypothesis that the

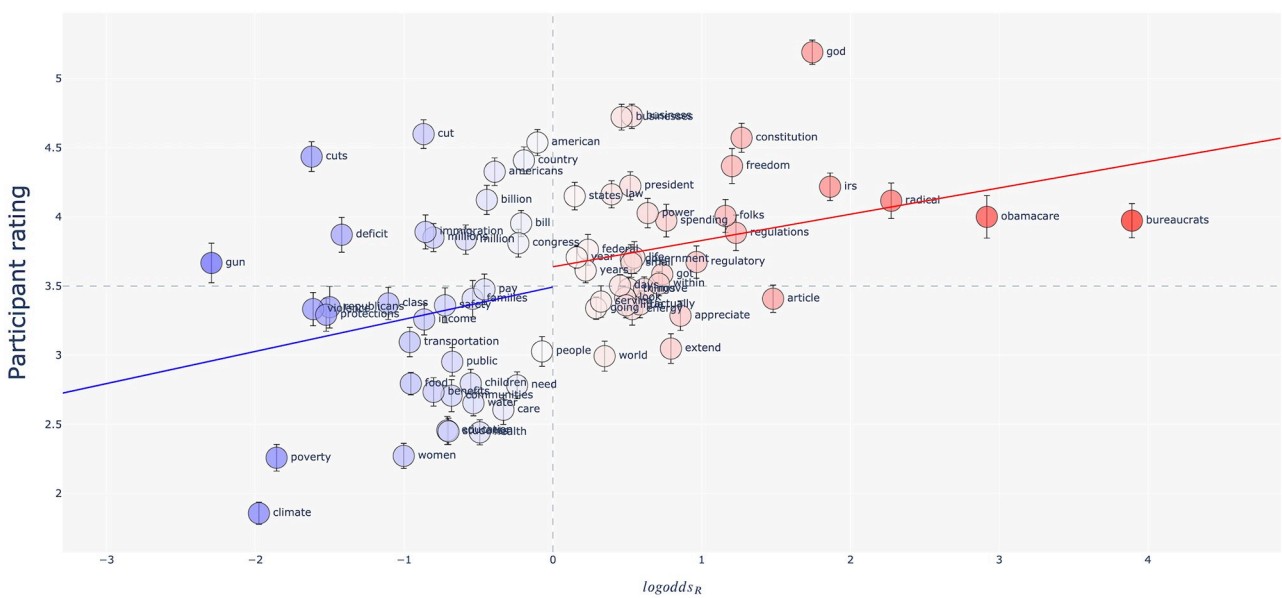

**Fig 2. Results of Study 1.** The $logodds_R$ of each word against the average rating given to the word. A rating of 6 corresponds to a judgment of "I am almost certain the speaker is a Republican," while a rating of 1 corresponds to "I am almost certain the speaker is a Democrat." Words are colored by $logodds_R$. Vertical bars mark one standard error around the mean of the ratings. The blue and red lines show the estimated linear trends using only the Democratic items and Republican items, respectively.

Republican words were as likely to be ranked above the Democratic words as to be ranked below. The $U$ statistic was 416 ($p < .01$), leading us to reject the null hypothesis.

In summary, the item-level analyses reaffirmed our conclusion that participant judgments aligned with the direction of politically conditioned variation more often than chance. Fig 2 shows the correspondence between the $logodds_R$ of each stimulus and its corresponding average participant rating. In general, words with a higher $logodds_R$—words that are more likely to be spoken by a Republican—tend to be *judged* as more likely be spoken by a Republican.

**Participant-level analyses.** On average, participants accurately classified 58.62% ($SE = .59\%$) of the items (or about 44 out of 75 words), with most participants (127 out of 147) performing better than chance. This number was significantly different from chance performance ($t_{146} = 14.66$; $p < .01$). They classified 57.52% ($SE = 1.10\%$) of the Democratic words correctly, and 59.69% ($SE = .91\%$) of the Republican words correctly. Both of these percentages were significantly higher than chance performance.

We also defined a measure of participant-level discriminability as the Cohen's $d$ between each participant's judgments on the Republican stimuli and their judgments on the Democratic stimuli. Recall that higher ratings indicate that the participant judged the speaker as more likely to be a Republican, so a participant with a positive Cohen's $d$ tended to make judgments in the right direction. The mean individual-level Cohen's $d$ is .40 ($SE = .03$), which is significantly higher than 0 ($t_{146} = 15.56$; $p < .01$).

## Discussion

From Study 1, we concluded that participants' judgments aligned with the direction of politically conditioned variation, and that this effect held for most participants and most items. However, recall that Study 1 did *not* control for our main theoretical confound: word *sense*. Studies 2 and 3 examine whether the alignment between human judgment and politically conditioned variation holds even when word *sense* is controlled for.

## Study 2: Controlling for word *sense* using cosine similarity

In Study 2 we test whether the sensitivity to politically conditioned variation we observed in Study 1 persists when word *sense* is held constant. We therefore wanted participants to make choices between pairs of words whose corresponding *senses* were as close as possible. But how does one measure the closeness of two words' *senses*?

Work in linguistics and cognitive science suggests that a word's context is an important contributor to its meaning: If words *A* and *B* appear near sets of words $C_A$ and $C_B$ respectively, *A* and *B* have similar *senses* when $C_A$ and $C_B$ are very similar [57, 58]. Consider the example given in the introduction: While the words "financial" and "monetary" may not be used concurrently (the phrase "financial monetary policy" sounds redundant and quite awkward), both words will likely co-occur with words like "policy," "markets," and "banks." This perspective can be summed up in a famous quote by Firth (1957): "You shall know a word by the company it keeps" [59].

Distributional semantics (DS) models are computational instantiations of this perspective: DS models build representations of word meaning on the basis of information about how the words co-occur in natural language data [60]. One popular DS model is word2vec, which projects each word of its input into a common, lower-dimensional space [61, 62]. Words that are closer together in this space are more semantically similar [63, 64]. One common way to measure this distance is using *cosine similarity*, or the cosine of the angle between the vector representations of two words. We therefore operationalized the word *sense* similarity of two words as their cosine similarity.

### Participants

175 subjects completed Study 2 on MTurk. After excluding 79 participants for failing our attention check, our analyzed sample contains 96 participants, including 50 self-identifying Democrats and 19 self-identifying Republicans. These exclusions do not affect our main results. Participants in the analyzed sample had a mean self-reported age of 40.68 ($SE = 1.12$), and included 37 men and 59 women. 80.21% of participants reported having voted in the 2016 presidential election. Participants completed the demographics questionnaire after the main survey.

### Method

We trained a word2vec algorithm using the software package Gensim on the Congressional Record corpus [65]. We took the 5% of words with the highest $PKL_D$, and the 5% of words with the highest $PKL_R$. For every pair of Democratic and Republican words in this set, we calculated the cosine similarity between the two words (662,596 pairwise comparisons in total). We then selected the 88 pairs with the highest cosine similarity (excluding pairs that contained words with proper names or acronyms that were not filtered out by the exclusion criteria listed in S1 Appendix). (We had predetermined that we wanted to present participants with 100 words pairs. Ten of the presented word pairs were included to address a separate research question, reported in Sloman, Oppenheimer, and DeDeo (under review) [66], and two were included as part of our attention check, detailed below. This meant 88 of the 100 pairs were left to directly test sensitivity to politically conditioned variation). It was possible that the highest cosine similarities among our restricted list of candidate pairs were not especially high in the context of the entire distribution of cosine similarities. To ensure that the constraint that each pair contain one highly Democratic word and one highly Republican word did not effectively impact the degree of *sense* similarity of the word pairs, we compared the cosine similarity of each item to the cosine similarities of 10,000 randomly selected word pairs from our corpus.

The least similar pair, *enable/encourage*, has a cosine similarity of.65, which is higher than 99.77% of this random sample of cosine similarities. We exclude responses on one pair from analysis, since the polarity of one of the words was not robust to later development of the list of stopwords (indicated in the list of stimuli in S2 Appendix). This was due to relative differences in the number of words excluded from Democratic and Republican speech (the total number of words spoken by Democrats and Republicans that were retained from the corpus after pre-processing are included in the calculation of $P(w|D)$ and $P(w|R)$, respectively). Assuming that cosine similarity is a successful proxy of word *sense* similarity, the set of stimuli presented to participants contains the 88 word pairs whose word *senses* are the most similar, subject to the constraint that each pair contains one highly Democratic word, and one highly Republican word.

Participants were asked to "[f]or each word pair, please guess which is indicative that the speaker is a Republican [Democrat]" (the full instructions are given in S2 Appendix). Each participant was randomly assigned to select either the word indicating that the speaker was a Republican, or the word indicating that the speaker was a Democrat. Both response and question order were randomized.

As before, two items were randomly reselected and included as attention checks. If, for either attention check question, a participant didn't respond to the question or didn't provide the same response as they had the other time they were presented with this pair, they were excluded from analysis.

## Results

Participants selected the word whose conditioned variation in usage aligned with their assigned condition 52.04% ($SE$ = 1.94%) of the time (an aligned response occurred when a participant asked to select the word more likely to be said by a Republican [Democrat] did indeed select the Republican [Democratic] word). While performance was slightly better than chance, a one-sided $t$-test against the null of 50% accuracy did not meet the conventional threshold for statistical significance ($t_{8351}$ = 1.05; $p$ = .15).

Further analysis showed that the effect was almost entirely driven by participants in the Democratic condition (i.e., participants who were asked to choose the word more likely to have been spoken by a Democrat). These participants ($n$ = 47) selected the Democratic word 53.71% ($SE$ = 2.04%) of the time, which was significantly higher than chance performance ($t_{4088}$ = 1.82; $p$ = .03). On the other hand, performance in the Republican condition ($n$ = 49) was slightly and not statistically different than chance ($\mu$ = 50.43%; $SE$ = 2.44%; $t_{4262}$ = .18; $p$ = .43).

To calculate a standardized effect size, we used the same method as for Study 1. We estimated a linear mixed model of participants' decisions to select the Democratic or Republican word (coded as a 0 and 1, respectively). The fixed effect was an indicator of whether the participant was in the Democratic or Republican condition (coded as a 0 and 1, respectively). Higher effect sizes thus indicate that selections align with the direction of conditioned variation. We specified participant-level intercepts as random effects (since the fixed effect did not vary within participants, we did not specify random participant-level slopes). We specified both item-level random intercepts and slopes. Using this model, we estimated a standardized effect size of.09. This reflects our interpretation of the results above: The effect of conditioned variation in Study 2 is in the hypothesized direction, but small and of questionable statistical reliability.

**Item-level analyses.**   We computed item-level accuracies by taking the mean of all participant judgments on each item. (Responses from participants in the Republican [Democratic]

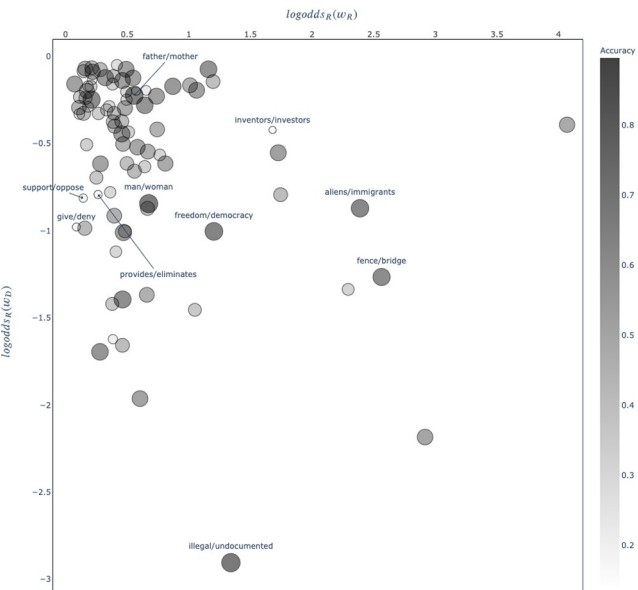

**Fig 3. Results of Study 2.** The $logodds_R$ of the Republican word against the $logodds_R$ of the Democratic word comprising each item. Markers are colored and sized by the mean participant accuracy. Large black dots indicate that when presented with this word pair, participants were usually able to identify which word was Republican and which word was Democratic. Small white dots indicate that when presented with this word pair, participants were systematically wrong about which word was Republican and which word was Democratic. The five pairs on which participants attained the highest average accuracy and the lowest average accuracy are labeled using the format *Republican word/Democratic word*. Highest accuracy: *illegal/undocumented*, *man/woman*, *father/mother*, *freedom/democracy* and *aliens/immigrants*. Lowest accuracy: *inventors/investors*, *give/deny*, *provides/eliminates*, *support/oppose* and *include/require*.

condition were coded as 1 if the word they selected was Republican [Democratic], and 0 otherwise). The mean item-level accuracy was 52.03% ($SE$ = 1.89%). In other words, participants selected the "correct" word (i.e. the word consistent with the direction of conditioned variation) for about 45 out of the 87 pairs. A one-sided $t$-test shows this is not significantly different than chance performance ($t_{86}$ = 1.07; $p$ = .14). Fig 3 shows the pairs participants classified most accurately, and the pairs on which participants systematically gave the wrong response.

Consistent with our overall finding above, performance was notably higher among participants in the Democratic condition. Among these participants, the average item-level accuracy was 53.67% ($SE$ = 2.00%; about 47 out of 87 word pairs), which was significantly different from chance ($t_{86}$ = 1.84; $p$ = .03). Among participants in the Republican condition, the average item-level accuracy was 50.44% ($SE$ = 2.37%), and was statistically indistinct from chance performance ($t_{86}$ = .19; $p$ = .43).

**Participant-level analyses.** We computed participant-level accuracies by taking the mean of participant responses across all items (responses were coded in the same way as for the item-level analyses). The mean participant-level accuracy was 52.03% ($SE$ = .68%; about 45 out of 87 word pairs). A one-sided $t$-test rejects the null of chance performance ($t_{95}$ = 3.00; $p$ <.01). We again found that participants in the Democratic condition performed significantly better than participants in the Republican condition: Participants in the Democrat condition had an average accuracy of 53.68% ($SE$ = .88%; about 47 out of 87 word pairs), which was significantly different than chance ($t_{46}$ = 4.17; $p$ <.01). Participants in the Republican condition had an average accuracy of 50.45% ($SE$ = .98%), which was statistically indistinct from chance performance ($t_{48}$ = .46; $p$ = .33).

## Discussion

In both Studies 1 and 2, participant responses aligned with the direction of politically conditioned variation more often than chance. However, in Study 2, when we attempted to control for word *sense*, the effect was small and not always significant. In particular, we found that participants asked to select the word more likely to have been spoken by a Republican performed marginally higher but not statistically differently from chance. One possibility was that this reflected the partisan skew of our sample: Our sample contained almost three times as many self-identifying Democrats as Republicans. If participants do indeed use the statistical distributions of language in their environment, a skewness in the statistics of that environment would lead to skewness in their cognitive representations of those distributions. Participants who are selectively exposed to speech from members of their own party may be better at recovering the signals from that party—i.e., Democrats may be particularly adept at recognizing Democratic words. However, when we broke down responses in the Democrat condition by self-reported party affiliation, we found that Republicans actually performed *better* than Democrats. Members of the two parties selected the correct word an average of 54.26% and 52.25% of the time, respectively.

Another possibility was that there were systematic differences in the degree of politically conditioned variation in the Democratic and Republican words. If the Democratic words exhibited a higher degree of conditioned variation, they could have been easier for participants to recognize. However, this was inconsistent with the fact that the mean $logodds_R$ value of the Republican words was actually *larger* in magnitude than the mean $logodds_R$ value of the Democratic words (.64 and -.58, respectively).

The asymmetry is puzzling to us, and we cannot rule out the possibility that it reflects a latent response bias that leads participants to select the Democratic word for reasons unrelated to politically conditioned variation. The way Democrats and Republicans speak likely differ in other systematic ways. For example, voters in urban areas are more likely to be Democrats [67], and we expect socially conditioned variation indicative of regional association often aligns with politically conditioned variation. A general bias to, e.g., select the word whose use or *sense* conveyed the speaker was from an urban area would result in apparently higher sensitivity to politically conditioned variation among participants in the Democratic condition.

Relatedly, a limitation of Study 2 is that our use of word2vec to control for word *sense* was not as effective as we had initially hoped. Many of the items used were not *sense* equivalents as we had intended. While word pairs with a high cosine similarity tend to be semantically related, they are not always synonymous. For example, many of the items were antonyms (e.g. *evening/morning* and *bad/good*) or words that both conveyed a quantity of something on different orders of magnitude (e.g. *billion/trillion* and *months/years*). Study 3 was designed to more rigorously establish *sense* equivalence between items. In addition, Study 3 incorporated the same task format as Study 1, which should eliminate any general response bias engendered by forced choice driving the results of Study 2.

## Study 3: Controlling for word *sense* using hand-coded synonyms

### Study 3a

**Participants.**  202 subjects completed Study 3a on MTurk. After excluding participants who failed a version of the instructional manipulation check [68], our analyzed sample includes 174 participants, including 77 self-identified Democrats and 45 Republicans. These exclusions do not affect our main results. Participants in the analyzed sample had a mean self-reported age of 40.35 ($SE$ = .87), and included 88 men and 85 women (1 participant reported

their gender identity as "other"). 84.80% of participants reported having voted in the 2016 presidential election. Participants completed the demographics questionnaire before the main survey.

**Methods.**   Our goal in Study 3a was to more completely isolate politically conditioned variation from variation in word *sense*. Because we wanted to ensure that the words we presented to participants were characterized by robust and generalizable politically conditioned variation, we restricted candidate stimuli to words for which the sign of $logodds_R$ was invariant to whether it was calculated using the Congressional Record or the presidential debates corpus (this step limited the possible stimuli to the 1,421 unique words that were present in both the Congressional Record and debates corpora, which is 8.76% of the original Congressional Record corpus).

We identified 530 Democratic words and 891 Republican words that are characterized by generalizable politically conditioned variation according to this criterion (37.30% and 62.70% of the candidate stimuli, respectively). We used these sets of words to construct pairs of *sense* equivalents: pairs of words that, as in Study 2, contained one Democratic word and one Republican word matched on *sense*. For Study 3, we relied on human coding of *sense* equivalence, detailed below, rather than the cosine similarity of the word pairs.

To extract *sense* equivalents, we manually sorted through this list of words, identified words which have familiar synonyms, and recorded these synonyms. To identify synonyms, one of the authors and a research assistant went through the list of words and consulted online resources, such as https://www.thesaurus.com/ and https://www.google.com/. We then further refined this list to only include words paired with synonyms that had the opposite partisan polarity in both corpora (in other words, if the target word was Republican, we only considered synonyms that were Democratic), and words that were not homonyms (excluding words like *run*, the meaning of which depends on if the person referred to is running a marathon, running a meeting, running out of time or running for office).

This left us with 26 pairs of *sense* equivalents in which one word was Republican and the other of which was Democratic (e.g. *kinds* (Democratic) vs. *types* (Republican) and *discussion* (Democratic) vs. *conversation* (Republican)). All participants saw one word from each pair, and rated those 26 words on the same scale used in Study 1. Based on the results of Studies 1 and 2, we chose the single-item rating format as opposed to the two-alternative forced choice format in order to maximize our power of detecting an effect. Another advantage of this task format is ecological validity: When we encounter words "in the wild," we usually make passing judgments, rather than forced choices.

We also gathered human ratings of the "substitutability" of each word. On the basis of pre-registered criteria (described in S3 Appendix), we excluded items whose constituents could not naturally replace each other in most ecological contexts. On this basis, we excluded one item from our analysis (*fear/terror*), leaving us with 25 items.

To further ensure the robustness of the direction of partisan signal in our items, we calculated 95% bootstrapped confidence intervals of the $logodds_R$ values of each word. To do this, we generated 100 artificial corpora. Each artificial corpus was created by randomly sampling speeches (with replacement) from the Congressional Record corpus. This allowed us to calculate 100 values of $logodds_R$ for each word, such that each value was calculated using data from a different artificial corpus. We determined 95% confidence intervals for each word as the interval within which the $logodds_R$ values corresponding to at least 95 of these artificial corpora fell. For 45 out of the 50 words included in our analyses, this interval did not cross 0 (the exceptions are *maintain* (a Democratic word), *types*, *employees*, *fundamental* and *gain* (Republican words)). For these words, we say that the direction of polarity is significant at the 95% level. Our results are robust even when the words with non-significant polarity are excluded from analysis.

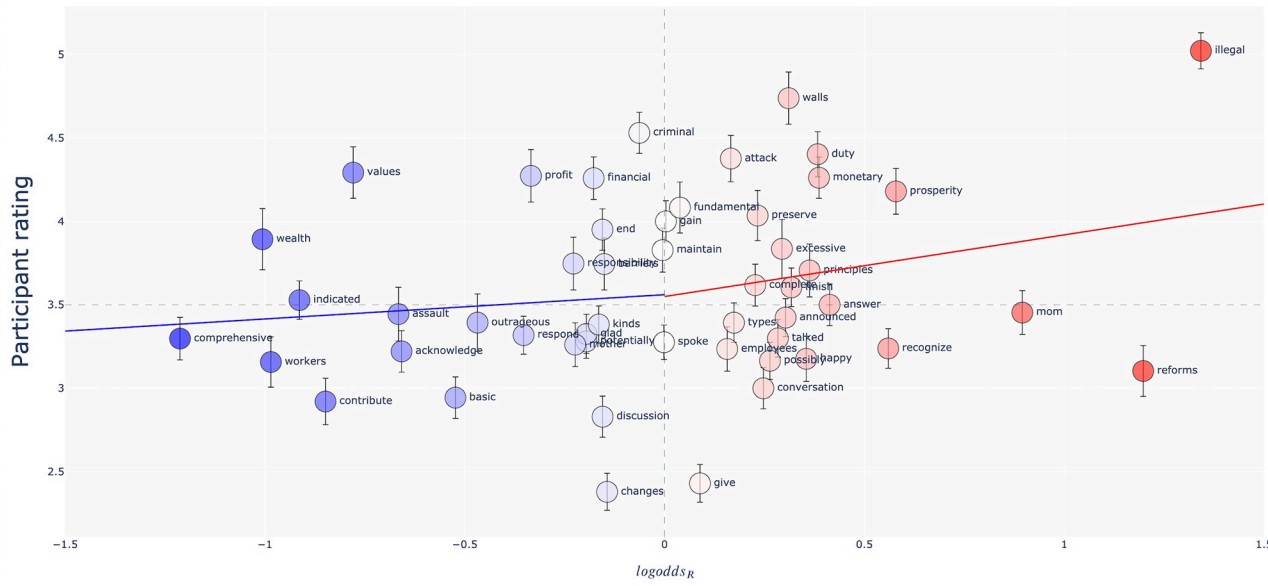

**Fig 4. Results of Study 3a.** The $logodds_R$ of each word against the average rating given to the word. A rating of 6 corresponds to a judgment of "I am almost certain the speaker is a Republican", while a rating of 1 corresponds to "I am almost certain the speaker is a Democrat." Words are colored by $logodds_R$. Vertical bars mark one standard error around the mean of the ratings. The blue and red lines show the estimated linear trends using only the Democratic items and Republican items, respectively.

Materials and pre-registrations associated with this study can be found at https://osf.io/x9qsp/.

**Results.** The mean rating on the Republican items was 3.70 ($SE$ = .12), just above the indifference point of the scale (recall that a higher rating corresponds to a judgment that the speaker is more likely to be a Republican). A one-sided $t$-test showed that this was significantly higher than the indifference point ($t_{2168}$ = 1.61; $p$ = .05). The mean rating on the Democratic items was 3.49 ($SE$ = .10), which was not significantly different than the indifference point ($t_{2173}$ = −.05; $p$ = .48). A two-sided $t$-test indicated that the difference between these two means was marginally significant ($t_{4341}$ = 1.29; $p$ = .10).

Using exactly the same method as for Study 1, we calculated a standardized effect size of .16. Because the design of Study 3a exactly mirrors the design in Study 1, we can directly compare these effect sizes: In Study 3a, which controls for word *sense*, we find a substantially smaller effect size than in Study 1.

This is suggestive but inconclusive evidence that participants are sensitive to the direction of politically conditioned variation. Fig 4 shows the correspondence between the $logodds_R$ value of each item and the average corresponding judgment collapsed across participants. In general, participants judge words as more likely to have been said by a Republican when the words *are* more likely to be said by a Republican.

*Item-level analyses.* Participants classified 29 out of the 50 items correctly (16 Democratic words and 13 Republican words). (As in our analyses of the data from Study 1, we consider an accurate classification to be when the direction of a participant's judgment with respect to the indifference point aligns with the direction of politically conditioned variation of the stimulus. Calculations of item-level accuracy use the same criterion but consider the average of all participants' ratings of each item). The average item-level accuracy was .53 ($SE$ = .03; i.e. about 53% of participants classified each item correctly), which was not significantly different from chance performance ($t_{49}$ = 1.18; $p$ = .12). Consistent with the asymmetry we noted above,

participants were slightly more accurate when only Republican items were considered ($\mu$ = .55; $SE$ = .04) than when only Democratic items were considered ($\mu$ = .52; $SE$ = .04; since there are only 25 items in each of these restricted sets, these samples are too small to run inferential tests on).

As for Study 1, we also ran a Mann-Whitney $U$ test. The $U$ statistic indicated that the overall rank order implied by participants' ratings was not distinct from the null ($U$ = 256; $p$ = .14).

If our effect was driven by attention to other characteristics, e.g. word *sense*, that happened to align with politically conditioned variation for a couple of critical items, our finding would not be strong evidence in favor of sensitivity to politically conditioned variation. In light of the strict criteria we applied to items for inclusion in Study 3a, the concern that our effect is driven by high performance on one or two items is particularly apt—especially given that our selection process saturated the set of candidate items, making it almost infeasible to address this potential concern in follow-up studies.

To see if this was the case, we calculated the average rating on each of the 50 items and looked at how these averages differed within the pre-specified pairs of *sense* equivalents. The complete results of this analysis are included as S1 File. In summary, while the mean rating on the Democratic word was lower than the mean rating on the Republican word for 18 out of the 25 word pairs, this difference was significant at the $p$ = .05 level for only eight of the 25 pairs (*comprehensive/complete, assault/attack, criminal/illegal, changes/reforms, responsibility/duty, barriers/walls, outrageous/excessive* and *basic/fundamental*). We found that the mean rating on the Democratic word was significantly *higher* than the mean rating on the Republican word for three of the 25 word pairs (*contribute/give, values/principles* and *end/finish*). The $p$-values associated with the remaining 14 word pairs fell between .05 and .95.

In the absence of an effect, we would expect the $p$-values associated with the word pairs to be uniformly-distributed. If, however, our data were unlikely under the null hypothesis, we would expect these $p$-values to be more skewed towards smaller values. S2 Fig shows the Q-Q plot of the $p$-values against the quantiles of the standard uniform distribution. Most of the plotted values fall below the 45˚ line, indicating that the density of the $p$-value distribution is concentrated around smaller values than the density of the standard uniform. A Kolmogorov-Smirnov test confirmed that the result that the cumulative density of $p$-values falls above the cumulative density of the uniform distribution is statistically significant ($D_{25}$ = .35, $p$ <.01). Despite the fact that the difference between the means of only eight of the word pairs was statistically significant, the distribution of effects at the item-level was unlikely to occur by chance. We interpret this as evidence against the possibility that the aggregate effect was driven by high performance on a couple of the word pairs.

*Participant-level analyses.* The mean participant-level accuracy was .53 ($SE$ = .01), with 108 participants (62.09%) performing better than chance. In other words, participants classified on average 28 out of the 50 items correctly. This was significantly higher than chance performance ($t_{173}$ = 4.73; $p$ <.01). The mean discriminability score across participants, defined exactly as for our analyses of Study 1, is .16 ($SE$ = .03), which is significantly different from 0 ($t_{173}$ = 4.95; $p$ <.01).

**Discussion.** Study 3a provides suggestive but inconclusive evidence that participants are sensitive to the direction of politically conditioned variation. Of note was the asymmetry we found between participants' sensitivity to variation in the direction of Republicans and variation in the direction of Democrats. In Study 2, we found that participants asked to select the word more likely to have been spoken by a Democrat were significantly more accurate than participants asked to select the word more likely to have been spoken by a Republican. In contrast, Study 3a finds that participants' judgments are better aligned with the direction of politically conditioned variation for Republican words. This suggests that if an unobserved response

bias was responsible for the asymmetry in alignment in Study 2, the design of Study 3a eliminated this bias.

But why did we observe a seemingly incongruent asymmetry in the alignment of judgments between the Republican and Democratic items in Study 3a? Our sample was again skewed towards Democrats, suggesting selective exposure to Republican speech was not responsible for this difference. We again considered the possibility that there were systematic differences in the degree of conditioned variation characterizing the Republican and Democratic items. The analyses in S4 Appendix and shown in Fig 4 show that judgments are positively correlated with the magnitude of the $logodds_R$ values of the items. However, the Republican items had an average $logodds_R$ value that was *smaller* in magnitude than the average $logodds_R$ value of the Democratic items (.38 vs. -.42). Therefore, sensitivity to the magnitude of the $logodds_R$ values could not explain the observed asymmetry.

Another speculative possibility is that participants in general had more exposure to Republican speech than Democratic speech. At the time of data collection in 2018, the Republican party had had majority representation in the House of Representatives, majority representation in the Senate and held the office of the presidency for the past two years [69]. More exposure may have enabled participants to more reliably encode conditioned variation indicative of Republican affiliation. In addition, dramatic shifts in rhetoric during the Trump administration [70] could have made instances of Republican speech more salient and easier for participants to retrieve from memory (see General discussion).

Overall, the statistical significance of the effect of the direction of conditioned variation was not robust to a variety of analysis strategies. This, combined with the small standardized effect size, suggested that the directional effect we found was fragile, a chance directional finding that would not replicate, or that Study 3a was not designed powerfully enough. Because of our stringent stimulus selection criteria, it is possible we were running up against a perceptual floor, and that participants were having a relatively harder time detecting weaker partisan signals. To minimize potential floor effects, in Study 3b we asked participants to make judgments about groups of words. We reasoned that exposing participants to several words with aligned patterns of politically conditioned variation simultaneously would result in a perceptual enhancement of the aggregate signal.

## Study 3b

**Participants.** 203 participants completed Study 3b on MTurk. After excluding participants who failed the same instructional manipulation check included in Study 3a, our analyzed sample contains 170 participants, including 78 self-identified Democrats and 37 self-identified Republicans. Participants in the analyzed sample had a mean self-reported age of 35.81 ($SE$ = .83). 102 men, 66 women, and two participants who identified as neither male nor female completed the study. 78.82% of participants reported having voted in the 2016 presidential election. Participants completed the demographics questionnaire before the main survey.

**Methods.** In Study 3b, we presented the 50 words we used in Study 3a in groups of five words each. Our rationale for this design was that the signals from the words would combine in such a way that we would overcome perceptual floor effects, increasing the power of our design to detect sensitivity to politically conditioned variation. Each word group contained exclusively Republican words or exclusively Democratic words. Each participant was presented with five groups composed of five words each, randomly generated in such a way that a participant was never presented with both an item and its *sense* equivalent. Participants were given the same instructions as in Studies 1 and 3a, with the exception that they were asked to imagine that they had overheard all the words in the list. The same version of the instructional

manipulation check was used to exclude participants for not paying attention. These exclusions do not affect our main results. Participants made a judgment about one word group per page, and questions auto-advanced after the participant selected their response.

**Results.** As in Studies 1 and 3a, higher ratings correspond to a higher perceived likelihood that the speaker is a Republican. The mean rating on the lists of Republican words was 3.96 ($SE$ = .07), while the mean rating on the lists of Democratic words was 3.54 ($SE$ = .06). (Standard errors are clustered at the participant level but not at the item level, since the study was designed such that exactly the same list of words was very unlikely to appear to more than one participant). As in our analysis of the data from Study 3a, we find that the mean judgment on the Republican items was significantly higher than the indifference point of 3.5 ($t_{417}$ = 6.95; $p$ <.01), but that the mean judgment on the Democratic items was not significantly different from the indifference point ($t_{431}$ = .67; $p$ = .75). A one-sided two-sample $t$-test showed that the mean judgment on the Republican items was significantly higher than the mean judgment on the Democratic items ($t_{848}$ = 4.51; $p$ <.01). Participants judged clusters of words as more likely to be spoken by a Republican when those words *were* more likely to be spoken by a Republican.

We used the same method as for Studies 1 and 3a to calculate an effect size, with the exception that we did not include item-level random effects. We calculated a standardized effect size of .30, which reflects that participants tended to rate lists of Republican words as more likely to have been said by a Republican than the individual words we presented in Study 3a.

*Participant-level analyses.* The mean participant-level accuracy, defined as for Studies 1 and 3a, was 57.18% ($SE$ = 1.58%), indicating that, on average, participants correctly classified about 3 in 5 word lists. 110 out of 170 participants performed better than chance. Consistent with the asymmetry in performance on the Republican and Democratic items noted above, performance differed dramatically in the two sets of items: The average participant-level accuracy on the Republican items was 64.71% ($SE$ = 2.19%), compared to only 49.80% ($SE$ = 2.17%) on the Democratic items.

We do not calculate discriminability scores as for Studies 1 and 3a. Since each participant saw only five word lists in total, we do not believe such a small sample provides reasonable estimates of the Cohen's $d$ between their judgments of Republican and Democratic lists.

## Discussion

Studies 3a and 3b demonstrate that even when word *sense* is controlled for, people are more likely to associate Republican language with Republicans. However, we found in both studies that participants did about as well as chance in associating Democratic language with Democrats. In our discussion of Study 3a, we speculated that that may reflect a skew in the amount of Republican vs. Democratic public-facing speech in participants' recent memories.

While Study 3 was designed to rule out the possibility that our results were due to inferences based on the *senses* of the words, our study design did not control for other attributes of words that have an impact on judgments of political affiliation. Namely, Sloman et al. (under review) show that the valence of language has a dramatic impact on such judgments [66]. The regression analyses in S4 Appendix show that when valence is controlled for, participants' judgments track the direction of *logodds$_R$* in both studies: Participants express more certainty that a word indicates the political affiliation of a speaker when that word exhibits stronger politically conditioned variation. Combined with the item-level analysis shown in S1 File, the findings of Study 3 are consistent with a sensitivity to politically conditioned variation that is more pronounced in the set of Republican words we identified (see again Fig 4). Taken together with the results of Studies 1 and 2, our evidence suggests listeners are sensitive to

politically conditioned variation. Importantly, both the magnitude of our effects and systematic asymmetries in participants' ability to recover variation were context-dependent. While our work finds that politically conditioned variation is *one* cue that enters participants' judgments, it has exposed rather than filled gaps in our knowledge of the large range of other cues on which people rely when using language to make inferences about a speaker's identity. We hope that future work can shed more light on the differences between the results from each of our studies.

## General discussion

Our results show that participants tend to classify hypothetical speakers as Democrats or Republicans in a way that reflects politically conditioned linguistic variation. This is consistent with our hypothesis that people can access and use politically conditioned variation in language when making judgments about a person's political identity. As we discussed above, conditioned variation in speech patterns emerges across many different demographic dimensions. To the best of our knowledge, it has remained until now an open question whether or not people are able to recover patterns in variation along the dimension of political identity. Combined with foundational results in categorical perception and perceptual learning [71, 72], our findings suggest that speakers' political identity is at least a somewhat salient perceptual category to the respective listeners. This has implications for a deeper understanding of the dimensions that contribute to people's representations of others, and how these higher-level representations interact with low-level perceptual and learning processes. S5 Appendix presents a brief exploration of some potential contributors to this feedback cycle. While those results are inconclusive, we believe the questions raised remain a promising avenue for future research.

One of our contributions is to the development of methods to explicitly disentangle conditioned variation from other forms of information that learners absorb from language. Establishing external validity is an especial challenge for researchers interested in behavioral responses to the statistics of language. Socially conditioned variation is a difficult construct to operationalize in an ecologically valid way, when its "ecology" is as varied as the number of distinct languages, contexts and speech patterns. Often, work on statistical language learning uses methods such as generating artificial grammars [2, 13, 73–76], simulation [77] or in-depth analyses of the speech patterns of a particular community or demographic group [4, 5, 15], all of which compromise some degree of external validity to achieve internal validity. As discussed above, and especially in our discussion of Study 3, our work by no means eliminates this challenge. However, we consider our approach an incremental step forward for a field interested in exploring ways of creating more ecologically-valid experimental stimuli. We along with Preoțiuc-Pietro et al. (2016) [6] exploit the availability of powerful techniques to mine large-scale data sets and recover the statistics of natural language. Performing our analysis at the word level allowed us to both present subjects with meaningful ecological units, and identify and control for plausible mediators.

While we believe our methods are a contribution to identifying socially conditioned variation outside the lab, our work nevertheless faces limitations to ecological validity. Participants in our studies made judgments about isolated, decontextualized words. However, speakers almost always produce sentences or paragraphs at a time. We expect politically conditioned variation also operates at the level of phrases [30] and sentences (e.g. "build the wall;" see discussion below and [78]). While extending our approach to phrases and sentences is an obvious avenue for future work, we chose not to do so because of the additional difficulties it would introduce in controlling for *sense* divergence. To the extent that phrases imply a larger network

of semantically related conceptual structures than isolated words, it would be an exponentially more difficult task to identify pairs of phrases with opposing directions of politically conditioned variation which were also matched on every element of their respectively encoded semantic networks.

A related limitation to the ecological validity of our results is our reliance on a controlled experimental setting. While this was necessary to isolate and manipulate measures of socially conditioned variation and intuitive judgments, it differs from contexts in which our participants encounter political speech "in the wild" in important ways. It is possible that our effects would be diluted or even eliminated in more ecologically valid settings, where decision-makers have a rich set of contextual and semantic information to draw from. However, socially conditioned variation and word *sense* often align and may interact in unexpected ways. For example, word *sense* could reinforce and contribute to the formation of more contiguous and salient episodic linguistic representations, rendering the conditional statistical distributions more accessible and usable. For example, consider a word pair on which participants in Study 3a did especially well: *barriers* (a Democratic word) vs. *walls* (a Republican word). One of Donald Trump's policy proposals was to build a barricade on the U.S.–Mexico border [78]. While the structures that exist and which the Trump administration planned to build along the U.S.–Mexico border are as accurately described as *barriers* as *walls* (if not more accurately [79]), his campaign came to be associated with the phrase "build the wall" [78]. While the concepts evoked by the proposal were very different from the concepts evoked by the Democrats' proposed immigration reforms, participants relying on only the *sense* of *barrier* and *wall* would not have been able to distinguish one or the other as more likely to be referring to Trump's proposal. Rather, we speculate that the starkness of the conceptual divergence between the two parties' proposed policies made the phrase "build the wall" more salient to participants, making it easier to encode the conditional distributions associated with the word *wall*.

Of course, we cannot completely rule out the possibility that our stimuli in Study 3 were not perfect *sense* equivalents. For example, one of our words pairs consists of *wealth* (a Democratic word) and *prosperity* (a Republican word). The Merriam Webster dictionary defines *wealth* as "abundance of valuable material possessions or resources" (first of four definitions [80]) and *prosperity* as "the condition of being successful or thriving" [81]. While the words have extremely similar meanings, one might think of forms of prosperity, such as social or intellectual fulfillment, as less similar to their concept of wealth. It's possible that a difference in responses on this item could have be driven not by attention to politically conditioned variation, but by an association between participants' concepts of a Republican and non-monetary forms of prosperity (while participants on average correctly guessed that *prosperity* is more likely to be said by a Republican, this difference was not significant at the $p = .05$ level). As discussed in our explanation of the methods for Study 3a, the stimuli used in Study 3 were the only word pairs that met our criteria for *sense* equivalence out of an initial pool of 1,421 words. Using the method described in S3 Appendix, we also ensured each word was exchangeable in context for its *sense* equivalent. We believe this maximized the semantic alignment within each word pair. While we cannot completely rule out the possibility that the words comprising each pair yielded different conceptual information, our data show that participants' judgments align with politically conditioned variation even when we assured there were minimal differences in the *senses* of corresponding words.

Finally, and as we discuss above, the corpus on which we based our operationalization of politically conditioned variation, the Congressional Record, may not be representative of the distribution of political speech to which our participants had been exposed. As we mention above, we chose to use the Congressional Record as the basis of our measurements of politically conditioned variation in favor of, e.g., transcripts of the presidential debates because it is

considerably larger and thus has the potential to provide more precise estimates for more words, and because the amount of Democratic and Republican speech is roughly balanced, making it less likely bias or lack of precision in our estimates would differ systematically by party. However, future work could attempt to replicate or extend our findings using corpora that contain more familiar, public-facing language.

## Mechanisms of politically conditioned variation

Above we showed that there is measurable between-party statistical variation in the use of language, variation that can be used to predict the party affiliations of speakers in other contexts. But what drives this systematic divergence? Are Democratic politicians intentionally trying to promote "discussion," while Republicans explicitly encourage "conversation"?

In politics, differentiating word use is often strategic. For example, beginning in the 1990s Frank Luntz began to put forth suggestions for specific terminologies Republicans could adopt to influence how voters thought about different issues (e.g. by referring to "tax simplification" instead of "tax reform;" [82, 83]). While tactical linguistic shifts such as those suggested by Luntz are intended to influence voters' higher-order representations of party-line issues, linguistic differences could result from strategic word choice even if the speaker does not invoke a specific semantic framing. Smaldino, Flamson, and McElreath (2018) discuss the dynamics of *covert signaling*, the intentional broadcasting of signals that convey information to in-group members, but are ambiguous enough to avoid offending or stirring conflict with out-group members [84]. Engaging in such "dog-whistle politics" allows politicians to make allusions to controversial stances that escape the awareness of members of their ideological out-group [85–87].

Diermeier et al. (2012) speculate that such coded appeals could partially explain variation in how the two parties use *sense* equivalents [29]. In order to convey a strong ideological stance to their base without alienating more moderate voters, they suggest that politicians may rely on the signaling power of the speaker's selection among these *sense* equivalents:

> For example, among the separating adjectives for Democrats we find the word *gay*, and for the Republicans we find the word *homosexual*. In other words, the correct use of terms signals one's political 'type' to constituencies that care a great deal about these issues. [29]

Alternatively, differences in speech patterns could reflect other latent differences in demographic attributes, personality traits or cognitive style of the members of the two parties. Extensions of our research could involve exploring differences in the degree of analytic thinking conveyed by the language of the two parties (e.g. [88]) or differences in the language used in support of vs. opposition to proposed policies (e.g. [89]).

While acts of speech production are the direct generating causes of the natural language data we analyzed, politicians' choices of words are also reflective of the language listening and learning processes they themselves have engaged in. Importantly, conditioned variation can also emerge organically from implicit language learning mechanisms. The iterative compounding of individual-level biases during language learning and transmission can result in systematic linguistic divergence [7, 73–76, 90]. We speculate that the use of glottal stops by younger Scots [8] and the term "yinz" by Pittsburghers [11] does not reflect deliberate, conscious attempts to signal identity, but rather unconscious patterns driven by, e.g., selective exposure on the part of language learners. Implicit learning mechanisms could similarly explain linguistic divergences among Democrats and Republicans.

The true drivers of politically conditioned variation are almost certainly a complex and dynamic combination of strategic word choice, latent conditioning variables and implicit

learning mechanisms—in addition to other factors we have not mentioned. In light of researchers' unprecedented access to natural language data and computing power, we are optimistic that future work can help construct a more complete picture of the conditioning process. Crucially, regardless of the mechanisms driving socially conditioned variation, the statistical information that emerges is a valid cue of the speaker's group membership. Our work contributes to a growing body of literature that people's judgments do, in fact, reflect this information [2, 6, 15, 16].

## Analog vs. discrete signals

Although many of our analyses made use of the magnitude of the *logodds$_R$* values, our interpretations of our results usually only considered behavioral correspondence with the *direction* of the conditional statistics: whether or not the word was categorized as Democratic or Republican. Prior work suggests that people's ability to categorize language-related stimuli on the basis of differences in perceptual characteristics, including differences in relative frequencies, reflects some amount of sensitivity to the magnitude of these differences [15, 35, 71]. Indeed, some of our analyses pointed to some such correspondence (see, e.g., Figs 2 and 4 and S4 Appendix). Future work could further investigate the degree of precision with which we encode and respond to identity-related linguistic input.

## Consequences of sensitivity to politically conditioned variation

To the extent that a person's political affiliation is an important predictor of their beliefs, behavior and interests, sensitivity to politically conditioned variation could allow listeners to infer characteristics about a speaker [23, 24]. Knowing the cues we use to infer partisan identity could inform an understanding of the bases on which listeners form implicit judgments of speakers. Given that even valid cues are probabilistic, listeners' judgments are likely often inaccurate. More awareness of the cues we rely on may help us recognize when our inferences are based on indirect cues like linguistic variation, rather than on confirmed and explicitly relevant information.

Language is perhaps the most important vehicle by which we convey information to others. Much of modern-day political discourse takes place over social media, online articles and email, where writers lack communicative mechanisms such as body language and tone of voice. In this context, we speculate that word choice is an especially important source of information. Linguistic variation is an important tool in the conveyance of partisan messaging—and understanding sensitivity to it can help us better understand receptiveness to such messaging. For example, the success of tactics like covert signaling and dog-whistle politics relies on listeners and readers being able to pick up on strategic linguistic variation. Cues to partisanship likely serve as more than just indicators of ideological alignment. While we may feel more ideologically similar to our political in-group members, we may also attribute other positive characteristics to them, such as trustworthiness: Word choice could lend source credibility to a speaker, perhaps leading us to believe or discredit information that is difficult for us to independently verify.

## Conclusion

We used natural language processing techniques to identify politically conditioned variation in public-facing U.S. political speech. We then demonstrated that human judgments align to some extent with this variation, even when cues such as word *sense* are controlled for. We contribute to a body of work examining the conditions under which people are sensitive to socially conditioned variation [2, 6, 15, 16]. Importantly, in some of our study designs the degree of

alignment was small, highlighting that more work needs to be done to more fully triangulate these conditions. Overall, our work shows that political party affiliation is a meaningful dimension along which people have the ability to track not only variation between overt value systems and policy preferences, but also subtle variation between distributions of word usage.

## Supporting information

**S1 Fig. Correlation between *logodds_R* values calculated using the Congressional Record and *logodds_R* values calculated using the presidential debates as a function of degree of politically conditioned variation.** The *x*-axis indicates deciles of the distribution of *logodds_R* values (calculated using the Congressional Record) of the 2,408 words that appear in both the Congressional Record and presidential debates corpora. Words in the leftmost bins are highly Democratic (have a very low corresponding *logodds_R*), while words in the rightmost bins are highly Republican (have a very high corresponding *logodds_R*). On the *y*-axis are the Pearson's correlation coefficients between the *logodds_R* values calculated using the Congressional Record and the presidential debates for the words that fall in each bin. For example, when the sample is restricted to the 241 words whose corresponding *logodds_R* was above the 10% and below the 20% cut points, the correlation is.18. The U-shape indicates that this correlation is highest for words with higher absolute values of *logodds_R*. The highest correlation (.40) occurs for words in the 10% decile (the most Democratic words). The lowest correlation (.02) occurs for words in the 60% decile.
(PNG)

**S2 Fig. Q-Q plot of theoretical quantiles of a standard uniform distribution against *p*-values in S1 File.** The closer the points fall to the red diagonal line, the more the distribution of *p*-values resembles what would be expected under the null hypothesis. The observed pattern shows that most *p*-values are smaller than would be expected under the null hypothesis, although the highest *p*-values, which correspond to the items that show a significant effect in the *opposite* direction of our hypothesis (see caption for S1 File), are higher than would be expected under the null.
(PNG)

**S1 File. Item-level results from Study 3a.** Table displaying word-level results from Study 3a and item-level *p*-values. For eight word pairs, the Republican word was rated as more likely to have been said by a Republican at the *p* <.05 level (the pattern predicted by our hypothesis): *comprehensive/complete*, *assault/attack*, *criminal/illegal*, *changes/reform*, *responsibility/duty*, *barriers/walls*, *outrageous/excessive* and *basic/fundamental*. For three word pairs, the Republican word was rated as *less* likely to have been said by a Republican at the *p* <.05 level (the opposite of the pattern predicted by our hypothesis): *contribute/give*, *values/principles* and *end/finish*. Out of the 14 items that did not show a significant difference in either direction, ten exhibited a difference directionally consistent with our hypothesis.
(XLSX)

**S1 Appendix.**
(PDF)

**S2 Appendix.**
(PDF)

**S3 Appendix.**
(PDF)

**S4 Appendix.**
(PDF)

**S5 Appendix.**
(PDF)

# Acknowledgments

We are grateful to Stephanie Rifai, Robert Feinstein, Nicholas Cardamone and Lauryn Patt for help with various aspects of this research project, and to Nicole Oppenheimer for help proof-reading. We are also grateful to anonymous reviewers for helpful comments and feedback.

# Author Contributions

**Conceptualization:** Daniel M. Oppenheimer, Simon DeDeo.

**Data curation:** Sabina J. Sloman.

**Formal analysis:** Sabina J. Sloman.

**Investigation:** Sabina J. Sloman, Daniel M. Oppenheimer, Simon DeDeo.

**Methodology:** Sabina J. Sloman, Daniel M. Oppenheimer, Simon DeDeo.

**Project administration:** Sabina J. Sloman, Daniel M. Oppenheimer, Simon DeDeo.

**Resources:** Daniel M. Oppenheimer, Simon DeDeo.

**Supervision:** Daniel M. Oppenheimer, Simon DeDeo.

**Validation:** Sabina J. Sloman.

**Visualization:** Sabina J. Sloman.

**Writing – original draft:** Sabina J. Sloman, Daniel M. Oppenheimer, Simon DeDeo.

**Writing – review & editing:** Sabina J. Sloman, Daniel M. Oppenheimer, Simon DeDeo.

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
