## [Decision Letter · Decision Letter 0]

4 Nov 2020

PONE-D-20-30828

Can we detect conditioned variation in political speech? Two kinds of discussion and types of conversation

PLOS ONE

Dear Dr. Sloman,

Thank you for submitting your manuscript to PLOS ONE. After careful consideration, we feel that it has merit but does not fully meet PLOS ONE’s publication criteria as it currently stands. Therefore, we invite you to submit a revised version of the manuscript that addresses the points raised during the review process.

Although both reviewers were positive about your manuscript, they raised a number of important issues that should be considered in a revision.  I summarize the main points below, but if you do submit a revision you should address all of the reviewers' comments.

First, your main argument hinges on the distinction between semantic and statistical information.  As reviewer 2 notes, however, that distinction is not clear-cut and some approaches equate the two. So this needs to be clarified and I think editing and changes in terminology (as suggested by Reviewer 2) is probably the way to go, although there may be other possibilities.

Second, the description of the methods and the reporting of the results requires a major overall.  In many places it is difficult to tell how the analyses were conducted and in other places it is unclear what the results mean.  For example when you say mixed model you should provide more information regarding the nature of that model (e.g., what were the random and fixed effects?). And for all reported effects you should follow APA guidelines and report the df and effect sizes. Reviewer 2 has multiple, detailed comments in this regard and you should respond to all of them.

Third, both reviewers encourage greater contextualization of your work and encourage additional speculation regarding the meaning of your findings, especially in the context of single words vs. phrases and sentences. Although this is less critical, these comments do need to be addressed.

A rebuttal letter that responds to each point raised by the academic editor and reviewer(s). You should upload this letter as a separate file labeled 'Response to Reviewers'.A marked-up copy of your manuscript that highlights changes made to the original version. You should upload this as a separate file labeled 'Revised Manuscript with Track Changes'.An unmarked version of your revised paper without tracked changes. You should upload this as a separate file labeled 'Manuscript'

We look forward to receiving your revised manuscript.

Kind regards,

Thomas Holtgraves, Ph.D.

Academic Editor

PLOS ONE

Journal Requirements:

2. Please note that according to our submission guidelines (http://journals.plos.org/plosone/s/submission-guidelines), outmoded terms and potentially stigmatizing labels should be changed to more current, acceptable terminology. In order to avoid conflation between gender and sex, "female” or "male" should be changed to "woman” or "man" as appropriate, when used as a noun.

3. Please upload a new copy of Figure 3 as the detail is not clear. Please follow the link for more information: " ext-link-type="uri" xlink:type="simple">https://blogs.plos.org/plos/2019/06/looking-good-tips-for-creating-your-plos-figures-graphics/"

4. We noted in your submission details that a portion of your manuscript may have been presented or published elsewhere.

"No. The authors have a separate project on how people respond to the valence of political speech, which is cited in the submitted manuscript as [38] Sloman SJ, Oppenheimer D, DeDeo S. One Fee, Two Fees; Red Fee, Blue Fee: People Use the Valence of Others’ Speech in Social Relational Judgments; in prep. Some data for this paper were collected simultaneously with data for the accompanying manuscript, and are available at the same public repository. Because the research questions are distinct and the stimuli were analyzed separately we do not consider this a related manuscipt, but are happy to provide the working paper to editors upon request."

Reviewers' comments:

Reviewer's Responses to Questions

**Comments to the Author**

1. Is the manuscript technically sound, and do the data support the conclusions?

Reviewer #1: Yes

Reviewer #2: Partly

2. Has the statistical analysis been performed appropriately and rigorously? 

Reviewer #1: Yes

Reviewer #2: I Don't Know

3. Have the authors made all data underlying the findings in their manuscript fully available?

Reviewer #1: Yes

Reviewer #2: Yes

4. Is the manuscript presented in an intelligible fashion and written in standard English?

Reviewer #1: Yes

Reviewer #2: Yes

5. Review Comments to the Author

Reviewer #1: This paper presents interesting work how even subtle linguistic cues can be informative of a speaker’s political ideology. While participants perform better than chance at assessing a speaker’s party identity, effects are generally quite small although significant. This may be expected given that these studies use as stimuli single words that are isolated from any other cues or context.

In all four studies, the U.S. Congressional Record is used to identify words that may be diagnostic of a speaker’s party identity. However, Congressional speeches are only one form of political language, and probably the one that many people are not exposed to. To their credit, the authors compare against presidential debate corpora, but only 1,421 of the 2,408 of the polarized words that overlap between the Congressional Record and presidential debates have the same estimated polarity. While they manage to find effects even with this limited overlap, it would be informative to look at sensitivity to these statistical signals across different contexts (e.g. news media, social media, or even looking at public debate data vs. Congressional speeches).

In the introduction, “gun laws” vs. “gun control” is one example of how political identity might be inferred from language. However, the studies here consider only single words. Looking at bigrams or trigrams might be informative here, allowing for examples like the above or “tax reform” vs. “tax simplification”, mentioned in the discussion, to be included. My understanding of semantic equivalence is murky here, since these bigrams refer to the same policies, though the individual words “laws” and “control” clearly convey different semantic information. Given that the authors write that “gun control” and “gun laws” “contain the same semantic information” (line 37) I assume these sorts of phrases would be considered semantically equivalent. It may also be interesting to consider whether words that are linked to policies are particularly informative. On a broader level – is there some higher-order factor that connects words that are indicative of political affiliation? Whether that’s words referring to policies, something about complexity/analytic thinking (see Jordan Pennebaker, 2017), or something else entirely.

While I think the authors successfully illustrate their hypothesis that meaningful speech variation by political party exists, a potential next step could be looking at how different terms may be informative in different time periods. One of the item pairings that seemed particularly interesting was “wall”/“barrier” – this seems like it would only be indicative in the time of Trump’s campaign/election. The transcript analysis was limited to 2012-2017 – was there a particular reason for choosing this time frame? It seems important to explain this choice given that the results are likely sensitive to variations over time.

Reviewer #2: This paper reports the results of three studies (as well as a wealth of supporting analyses) to examine whether lay perceivers correctly discern whether given words are more frequently said by Democrats or by Republicans. I thoroughly enjoyed the paper and find it to bring an exciting approach to an area of widespread interest.

My main concern centers on the clarity and consistency of the methods – as I elaborate below, the reporting of the methods and results started out on a strong foot but quickly became quite complex in reporting. As a result, even with a background in these kinds of analyses, I was lost as to what exactly the authors were doing statistically to support their conclusions. This makes it difficult for me to evaluate whether the technical approach is sound. Even in Study 1, when the authors report “an ANOVA on a mixed model”, I had no context for this analysis: what was the mixed model? What was the baseline model that they were doing the ANOVA comparison against (presumably this was the model comparison approach, but again, it wasn’t clear to me).

Another point of confusion, for illustration: From the results presented on line 399, it seems the R word recoverability was greater than the recoverability for D words? Is this true? That is, it seemed that the R result was further from the midpoint than the D result. Separate one-sample t-tests comparing effect sizes would inform this interpretation. If it is true, then please offer a discussion or speculation as to why this might be the case.

Finally, a third major point of confusion in the results centered on the figures: although I appreciated the effort made to clearly describe the graph, I really struggled to wrap my head around exactly what was being visualized. More legends in the figure may be helpful. Please take a second pass at these visualizations to generate other possible ways of representing the results to make them more interpretable for the general audience of PlosOne. I found Figure 3 to be more interpretable than the others, so perhaps using that one as a model for the others will be helpful for the author.

A substantial revision to the writing will be required to streamline and sign post (i.e., using numbering or “first… second… third”, etc.) exactly what analyses are being done. Further, the authors may want to be much more systematic in the analyses they do for each study. As is typically in the behavioral sciences, I would recommend starting with descriptives and basic inferential tests (e.g., how many words were correctly classified, was that number significantly different from 0 from a one-sample t-test), and then moving on to one clear mixed-effects model that examines both item-level and person-level variance. The authors know their data much better than I (obviously!), so they will know exactly which model is most clear and compelling for their data. Present that model first and clearly (and consistently across studies), and then lay out the remaining additional analyses that probe secondary questions.

My second major suggestion concerns the terminology used. The authors spend a good amount of time in the introduction and throughout the manuscript emphasizing the dissociation between “statistical” and “semantic” information. But, unless we know the very precise definition of “statistical” that the authors are referring to, the more vague term of “statistical” makes this distinction hard to understand. If we use the lay concepts of “statistical” as referring to frequency in the environment, then “semantic” information can also be inferred from “statistical” information (indeed the “distributional hypothesis of semantics” from Firth and others would argue that semantic information is almost entirely driven by statistical information). My suggestion is (1) to really elaborate and emphasize the examples first and only after to point out that the exact same semantic concept is being represented in idiosyncratic ways (gun laws vs. gun control or you guys vs yinz); and (2) to consider alternative terminology. Statistical is perhaps too vague a term to capture this precise meaning. Is it better perhaps to say statistical dialect? Really it’s conditioned variation on the basis of group membership, so why not just use that terminology throughout (“conditioned variation over and above semantics” makes more sense to me than “statistics over and above semantics”)

Finally, I have a number of line-by-line small points where I was confused by the writing or results:

1. I was very confused in my first reading of the abstract - What does “direction of statistical information” mean? I don’t think a lay reader would understand this language since “statistical information” hasn’t been defined. Perhaps wording like “follows the trends of the ground-truth data” would be more clear? Alternatively, the paragraph starting on line 27 could be re-reported in a condensed form in the abstract to clarify my concern. The definitions in that paragraph were well described.

2. Line 53: Spell this point out for your reader - what exactly do you see as the clear relevance of this work?

3. For all the reported numbers of R and D words (e.g., lines 136-38), is the author merely reporting classification based on the SIGN of the effect size (not the MAGNITUDE)? Were these distributions of R and D words significantly different from a null distribution around zero? Throughout, I think it would be helpful to have more reporting of the distributions and magnitudes of effect sizes.

4. What does PKL stand for exactly? A sentence or two more is needed in this description. The log-odds description is great but then the author jumps into an altogether different approach (or so it seems); can the description be streamlined to only focus on the metric that is of most interest (and then the remaining metric is moved to supplemental?)

5. Lines 174: Please report proportions of words rather than exact raw values. 1421 words is how many of the original set?

6. Please justify the point of Study 2 much earlier - either by laying out the general program of studies and/or by including a short intro to the study before the methods. As I was reading through the methods of Study 2, I didn’t know the point of what I was reading. Similarly, it would be helpful to revise the title of this Study to be more informative (e.g., something referring to trying to disentangle the role of semantics and using cosine similarities to do so).

7. Line 287: How many pairwise comparisons was this in the end?

8. Line 320: One cannot “strongly” reject the null hypothesis in a Frequentist framework - if the author wants to discuss magnitude of evidence, I’d suggest going Bayesian…

9. Line 322: The lack of significance for item-level analyses here is likely because of limited power (assuming the author is aggregating at levels of items? This is not clear to me, see my comments above about clarifying your methods!).

10. Also, throughout, for all models and t-tests, please report df and effect sizes (e.g., Cohen’s ds), as per APA format

11. Why not look at sentence-level effects? These single words out of context are difficult to interpret. More justification is needed than what is given (i.e., that language learning and change occurs at the word level); the paper is not interested in language learning or change, so why is that justification relevant? The word level may be a good start, but it would be interesting to extrapolate beyond and think about the implications for speech patterns more generally. As the authors show, groups of words provide a much stronger signal than a single word; presumably this would be even more true if they were constructed as sentences or as paragraphs…

12. And finally, I’d love to see the author speculate a bit about downstream consequences of this effect on persuasion and influence? Is speech tailored to your own party more likely to persuade you? This seems particularly pertinent in a time of such political polarization and “fake news” across party lines - would be exciting to see this mentioned.

6. PLOS authors have the option to publish the peer review history of their article (what does this mean?). If published, this will include your full peer review and any attached files.

Reviewer #1: **Yes: **Nina Wang

Reviewer #2: No

---

## [Author Response · Author response to Decision Letter 0]

28 Dec 2020

The response below is also included in the rebuttal letter we have uploaded with our submission.

Dear Dr. Thomas Holtgraves,

Thank you for your careful consideration of and helpful feedback on our manuscript “Can we detect conditioned variation in political speech? Two kinds of discussion and types of conversation.” Our revised manuscript incorporates all the feedback provided by you and the two reviewers. Below, we respond to each point raised by you and the reviewers.

1. "First, your main argument hinges on the distinction between semantic and statistical information. As reviewer 2 notes, however, that distinction is not clear-cut and some approaches equate the two. So this needs to be clarified and I think editing and changes in terminology (as suggested by Reviewer 2) is probably the way to go, although there may be other possibilities."

Based on these comments, we have revised our terminology for distinguishing between statistical and semantic information. As Reviewer 2 rightly points out, the concept of semantic information is not well-defined, and our original terminology did not make it clear how it was distinct from statistical information. We hope our choice of the phrases “politically conditioned variation” (what our original manuscript referred to as “statistical information;” defined on lines 50--52) and “word sense” (what our original manuscript referred to as “semantic information;” defined on lines 28—29) make this distinction clearer.

2. "Second, the description of the methods and the reporting of the results requires a major overall. In many places it is difficult to tell how the analyses were conducted and in other places it is unclear what the results mean. For example when you say mixed model you should provide more information regarding the nature of that model (e.g., what were the random and fixed effects?). And for all reported effects you should follow APA guidelines and report the df and effect sizes. Reviewer 2 has multiple, detailed comments in this regard and you should respond to all of them."

As you and Reviewer 2 correctly point out, in our original manuscript we did not follow a standardized format for our results sections, did not break up our hypothesis tests by the partisan polarity of the stimuli, and did not report standardized effect sizes. Following helpful suggestions from Reviewer 2, we have broken up each of our results sections into subsections that detail our main analyses, participant-level analyses and item-level analyses, respectively. In the main analysis sections, we report a standardized effect size for each study, using the method of Rouder, Morey, Speckman, and Province (2012). Our reporting of the results of Studies 1 and 3 include the same analyses reported in the same order. While the design of Study 2 differs slightly, we follow the same analysis strategy and reporting format to the extent possible.

In addition to standardizing our analysis strategies and formatting, we have substantially revised our results sections in other ways:

• Based on Reviewer 2’s comments, we also include extensive analysis and discussion of differences in results when only Republican [Democratic] items are considered. 

• Based on Reviewer 2’s comments, we have included new figures that draw on what was Fig 3 in the original submission as inspiration.

• We have clustered standard errors used for our main analyses at the participant- and item-level.

• The tables in Appendices D and E contain slightly modified coefficient estimates that reflect minor changes to our model estimation strategies (using an L1 penalty in favor of OLS for the linear regressions in Appendix D and the inclusion of item-level effects in Appendix E). Most notably, in our original manuscript, we mentioned that a pre-registered regression model of the data from Study 3a (first column in Table 1) did not estimate a coefficient on logoddsR in the hypothesized direction. This was due to a programming error; when this is fixed, this model does show a coefficient estimate in the hypothesized direction. (None of the other changes affect our central theoretical claims.)

3. "Third, both reviewers encourage greater contextualization of your work and encourage additional speculation regarding the meaning of your findings, especially in the context of single words vs. phrases and sentences. Although this is less critical, these comments do need to be addressed."

There are a number of places where we have clarified our methods and included additional points of discussion based on feedback from both reviewers:

• On lines 790--797, we clarify our rationale for our choice to investigate politically conditioned variation at the level of single words rather than phrases or sentences. To summarize, this choice was in our judgment necessary to adequately control for differences in word sense between our stimuli.

• Both reviewers recommended we use a different example to highlight the distinction between politically conditioned variation and word sense. We have replaced the example of “gun control” vs. “gun laws” with “financial” (a Democratic word) vs. “monetary” (a Republican word; lines 27--44). We believe this addresses the concern raised by Reviewer 1 that the example included bigrams while our analyses occurred at the level of unigrams, and that our discussion on lines 27--34 addresses the concern raised by both reviewers that the example did not make salient our notion of semantic equivalence (i.e. sense equivalence). 

• In response to Reviewer’s 2 request that we speculate on the downstream consequences of sensitivity to politically conditioned variation, we include an additional section in our discussion entitled “Consequences of sensitivity to politically conditioned variation” (lines 904--926). Reviewer 2 noted that our findings could have important implications for political persuasion and the spread of fake news. We elaborate on this, providing additional discussion about how detection of politically conditioned variation may allow people to make additional inferences about the speaker, and, especially to the extent that it provides signals to ingroup or outgroup membership, could lead to differences in perceived trustworthiness and credibility.

• In lines 868—872, we include Reviewer 1’s insightful point that there are likely latent factors that reflect other differences between Republicans and Democrats, yet manifest themselves as politically conditioned variation.

• Reviewer 1 also notes that we did not justify our decision to use data from a five-year window of time to measure politically conditioned variation. We explain our rationale for this choice on lines 122--130. To summarize, we believe this time window achieves a balance between collecting enough data to recover reliable trends and our concern that a time window that extended back too far would overwhelm trends from more recent years.

Reviewer 2 also provided a number of additional line-by-line comments, each of which has been addressed in the revised manuscript:

1. We have altered the sentence in the abstract from “In a series of four studies, we demonstrate that participants’ judgments align with the direction of statistical information more often than chance, and that this effect persists even when potentially interfering cues such as the semantic content of the words are controlled for” to “In a series of four studies, we demonstrate that participants’ judgments track variation in word usage between the two parties more often than chance, and that this effect persists even when potentially interfering cues such as the meaning of the word are controlled for.”

2. On lines 63--75, we spell out the practical relevance of our work. To summarize, understanding sensitivity to politically conditioned variation can help us better understand the kinds and validity of cues we use to make implicit judgments about others.

3. We have attempted to clarify that Reviewer 2’s interpretation that classifications of words are based on the sign of their logoddsR value is correct by altering our phrasing on lines 162--163 to stress that classification depends on logoddsR being greater than or less than 0. Reviewer 2 also recommends we report and comment on the distribution of logoddsR values in the Congressional Record corpus. We have replaced Fig 1 (restating the important takeaways from this figure in the text; lines 188--196) with a figure visualizing this distribution. In addition, we address Reviewer 2’s question of whether this distribution was distinct from a plausible null distribution on lines 164--168.

4. As Reviewer 2 suggests, we re-structure the section “Measures of relative frequency” to focus on the logoddsR metric. We move discussion of PKL to lines 227--237, and clarify that “PKL” is an abbreviation for “partial Kullback-Leibler divergence.”

5. On lines 188 and 191, we report proportions as well as the absolute number of words that retain their polarity from the Congressional Record to the presidential debates corpora. 

6. We provide additional motivation for Study 2 before the corresponding “Participants” and “Methods” sections (lines 338--359).

7. We clarify that stimulus selection for Study 2 entailed 662,596 pairwise comparisons (line 373).

8. We remove the word “strongly” due to Reviewer 2’s observation that “strong rejection” cannot occur in a frequentist statistical framework.

9. Reviewer 2 suggests that the lack of a significant result for the item-level analysis of Study 2 (line 434) was likely due to a lack of power (since there were only 87 items). In the corresponding “Item-level analyses” section, we discuss additional analyses showing that performance was significantly higher than chance for participants in one of the two conditions. In light of other results from Study 2, we chose to stress the difference between conditions in discussion of the item-level result. However, we are happy to also speculate about the power of our tests if the reviewer believes it would aid readers’ interpretation of our results.

10. We have included degrees of freedom in reporting of all of our t-tests. While we believe Cohen’s ds are difficult to interpret for repeated measures data (see our discussion on lines 268--282), we provide a repeated measures analog developed by Rouder et al. (2012; see discussion above).

11. As mentioned above, we clarify our rationale for focusing on word-level effects on lines 782--794.

12. As mentioned above, we speculate on the consequences of sensitivity to politically conditioned variation in the section “Consequences of sensitivity to politically conditioned variation.”

Finally, we would like to respond here to your questions about overlap between the manuscript we have submitted to your journal (the present manuscript) and another manuscript of ours that is currently under review entitled “One Fee, Two Fees; Red Fee, Blue Fee: People Use the Valence of Others’ Speech in Social Relational Judgments.” The present manuscript addresses the research question “Can participants detect politically conditioned variation?” Sloman, Oppenheimer, and DeDeo (under review) addresses the research question “Do participants perceive the speech of members of their political in-group as more positive than the speech of members of their political out-group?” Addressing both research questions required analysis of participant judgments of the most likely political identity of a hypothetical speaker. We therefore found it convenient to ask participants in Studies 1 and 2 of the present manuscript to also make judgments about words that would test the research question addressed in Sloman et al. (under review). In Appendix B of the present manuscript, you and interested readers can see the full list of items participants in Studies 1 and 2 were presented with. We have clearly demarcated which of these items were included to test the research question addressed in the present manuscript, and which of these items were included to test the research question addressed in Sloman et al. (under review). These lists do not overlap; in other words, none of the data analyzed for and reported in the present manuscript is reported in Sloman et al. (under review). Since a) the research questions addressed in the present manuscript and in Sloman et al. (under review) are completely distinct, and b) none of the data analyzed for or reported in the present manuscript is reported in Sloman et al. (under review), we believe this does not constitute dual publication.

We hope our revisions adequately address your concerns about the clarity of our theoretical and methodological approach. We thank you and the reviewers again for the time taken to provide feedback on our work, and look forwards to your replies.

Sincerely,

Sabina J. Sloman

Dr. Daniel Oppenheimer

Dr. Simon DeDeo

References

Rouder, J.N., Morey, R.D., Speckman, P.L., Province, J.M. (2012). Default Bayes factors for ANOVA designs. Journal of Mathematical Psychology, 56(5), 356-374.

Jordan, K.N. Pennebaker, J.W. (2017). The Exception or the Rule: Using Words to Assess Analytic Thinking, Donald Trump, and the American Presidency. Translational Issues in Psychological Science, 3(3), 312-316.

Sloman, S.J., Oppenheimer, D., DeDeo, S. (under review). One Fee, Two Fees; Red Fee, Blue Fee: People Use the Valence of Others’ Speech in Social Relational Judgments.

---

## [Decision Letter · Decision Letter 1]

15 Jan 2021

PONE-D-20-30828R1

Can we detect conditioned variation in political speech? Two kinds of discussion and types of conversation

PLOS ONE

Dear Dr. Sloman,

I have read your revised manuscript (PONE-D-20-30828R1"Can we detect conditioned variation in political speech? Two kinds of discussion and types of conversation") as well as a review of the manuscript provided by one of the reviewers (Reviewer 2) of your original submission.  The reviewer and I both found your revision responsive to concerns raised in the initial round of reviews.  There are, however, a few additional tweaks to be made before your manuscript can be accepted for publication. First, please address all of the minor issues raised by Reviewer 2.  Second, because your samples consist of  MTurk workers I would like you to mention some of the limitations (e.g., non-naivety and trustworthiness) of using participants from this platform.  Third, for study 2, state in the method section that participants provided judgments of 100 word pairs, 12 of which were for a different study.  Fourth, please add a sentence within the manuscript that provides a reference to the location of your data and code (and please make sure that the studies in this manuscript are aligned with what is contained on your GitHub page).

That’s it.  Once you submit a revision I’ll review it and make a determination. Note that I won’t be sending this out again for review.

We look forward to receiving your revised manuscript.

Kind regards,

Thomas Holtgraves, Ph.D.

Academic Editor

PLOS ONE

Reviewers' comments:

Reviewer's Responses to Questions

**Comments to the Author**

1. If the authors have adequately addressed your comments raised in a previous round of review and you feel that this manuscript is now acceptable for publication, you may indicate that here to bypass the “Comments to the Author” section, enter your conflict of interest statement in the “Confidential to Editor” section, and submit your "Accept" recommendation.

Reviewer #2: (No Response)

2. Is the manuscript technically sound, and do the data support the conclusions?

Reviewer #2: Yes

3. Has the statistical analysis been performed appropriately and rigorously? 

Reviewer #2: Yes

4. Have the authors made all data underlying the findings in their manuscript fully available?

Reviewer #2: Yes

5. Is the manuscript presented in an intelligible fashion and written in standard English?

Reviewer #2: Yes

6. Review Comments to the Author

Reviewer #2: As a previous reviewer of this paper, I appreciated the authors attention to my earlier comments. Their extensive revisions have greatly strengthened the clarity of the methods, the clarity of the conceptual distinction between information of sense/conditioned variation in words (I agree with this terminology!), and the overall contribution of the work.

I have a few remaining minor comments for consideration:

1. I might have missed this the first time, but I was surprised by the fact that the correlation between log-odds for Congressional and Presidential speeches was only a small to medium correlation. Although it is significant (which can provide some validation of the method), it also points to the variation in Congressional vs. Presidential speeches. A single sentence or two elaborating on these differences would be helpful, either in the main text or SM. Where is this variation coming from?

2. Relatedly, which corpus is actually better reflection of the signals that lay perceivers are learning from? Presumably the Presidential speeches, as the authors themselves note. So why did the authors not use that as the primary corpus of interest for the authors initial generation of word stimuli? I understand the points the authors made to justify using the Congressional speeches, but if their goal is to look at lay perception of politically conditioned variation, then doesn’t it make sense to look at the most “lay” corpus for word differences? I also realize it’s impossible to go back and change this now, but it might help to address this in a footnote as a limitation or direction for future work.

3. Line 260 “to reject the null that this value was greater than or equal to…” – is this inclusion of “greater than” correct? It seems like you rejected the null that the value was equal to the indifference point. The same issue for the Democratic stimuli reported in the following paragraph. Please double check or clarify.

4. Thank you for the additional clarity about the mixed model and effect size estimation.

5. Line 373 – where did the choice of N = 88 pairs come from? Was there some principled cut-off for cosine similarities? Would choosing only the highest cosine similarity pairs weaken the effect even further? Such a finding would suggest that differences in word sense do, in fact, have some impact on the discrimination of R vs. D words.

6. Another alternative for why Republican discriminability may be higher (in addition to the frequency of exposure to speech as the authors mentioned) is that lay perceivers may have stronger stereotypes about Republicans than about Democrats. In other words, they may have clearer notions about the traits and cognitions of Republicans than Democrats. My intuition is that, especially during Trump’s presidency, lay perceivers see Republicans as a more extreme social group than Democrats. Another speculation to play with, perhaps?

7. The figures are all much improved!

7. PLOS authors have the option to publish the peer review history of their article (what does this mean?). If published, this will include your full peer review and any attached files.

Reviewer #2: No

---

## [Author Response · Author response to Decision Letter 1]

21 Jan 2021

The following comments are also included in the attached file entitled "Response to Reviewers."

Dear Dr. Thomas Holtgraves,

Thank you for your feedback on our manuscript “Can we detect conditioned variation in political speech? Two kinds of discussion and types of conversation.” Below, we point to where in our manuscript we have addressed each point you raised.

1. First, please address all of the minor issues raised by Reviewer 2.

The reviewer’s comments were (lettered rather than numbered to avoid confusion with your comments):

a. I might have missed this the first time, but I was surprised by the fact that the correlation between log-odds for Congressional and Presidential speeches was only a small to medium correlation. Although it is significant (which can provide some validation of the method), it also points to the variation in Congressional vs. Presidential speeches. A single sentence or two elaborating on these differences would be helpful, either in the main text or SM. Where is this variation coming from?

We agree with the reviewer’s insight that there is still much variation to be explained between the two corpora. While we cannot completely explain the sources of divergence, we include an additional supplementary figure, S1 Fig, that shows how the generalizability of politically conditioned variation varies with its strength in the Congressional Record corpus. We find that the strongest political signals are also the most generalizable. We summarize these findings on lines 197—202.

b. Relatedly, which corpus is actually better reflection of the signals that lay perceivers are learning from? Presumably the Presidential speeches, as the authors themselves note. So why did the authors not use that as the primary corpus of interest for the authors initial generation of word stimuli? I understand the points the authors made to justify using the Congressional speeches, but if their goal is to look at lay perception of politically conditioned variation, then doesn’t it make sense to look at the most “lay” corpus for word differences? I also realize it’s impossible to go back and change this now, but it might help to address this in a footnote as a limitation or direction for future work.

We address this limitation on lines 875—885 of our general discussion.

c. Line 260 “to reject the null that this value was greater than or equal to…” – is this inclusion of “greater than” correct? It seems like you rejected the null that the value was equal to the indifference point. The same issue for the Democratic stimuli reported in the following paragraph. Please double check or clarify.

The reviewer is correct to point out that this was a typo on our part. We have adjusted the sentence to read “A one-sided, one-sample t-test led us to reject the null that this value was less than or equal to the indifference point…” (lines 282—284). We hope this clarifies that we performed a one-sided test to test our alternative hypothesis that participants’ judgments on the Republican items would fall above the indifference point. We have also adjusted terminology when reporting results on the Democratic items (lines 289—290).

d. Thank you for the additional clarity about the mixed model and effect size estimation.

e. Line 373 – where did the choice of N = 88 pairs come from? Was there some principled cut-off for cosine similarities? Would choosing only the highest cosine similarity pairs weaken the effect even further? Such a finding would suggest that differences in word sense do, in fact, have some impact on the discrimination of R vs. D words.

On lines 401—406, we clarify that we predetermined that we would present participants with 100 items, ten of which were included to test the hypothesis discussed in Sloman, Oppenheimer, and DeDeo (under review) and two of which were included as part of our attention check. In other words, the choice of 88 pairs was an artifact of experimental constraints, not the result of a constraint imposed by the distribution of cosine similarities.

f. Another alternative for why Republican discriminability may be higher (in addition to the frequency of exposure to speech as the authors mentioned) is that lay perceivers may have stronger stereotypes about Republicans than about Democrats. In other words, they may have clearer notions about the traits and cognitions of Republicans than Democrats. My intuition is that, especially during Trump’s presidency, lay perceivers see Republicans as a more extreme social group than Democrats. Another speculation to play with, perhaps?

We interpret the reviewer’s comment as suggesting that participants’ cognitive representations of Republican speech were stronger and easier to access. We include this suggestion on lines 689—692, and believe it also links with the section of our general discussion on lines 838—854.

g. The figures are all much improved!

2. Second, because your samples consist of MTurk workers I would like you to mention some of the limitations (e.g., non-naivety and trustworthiness) of using participants from this platform.

On lines 215—229, we provide a brief discussion of the advantages and limitations of using MTurk as a recruiting tool, in addition to discussion of how these limitations would affect interpretation of our results.

3. Third, for study 2, state in the method section that participants provided judgments of 100 word pairs, 12 of which were for a different study.

As mentioned in our response to 1e, we clarify on lines 401--406 that we predetermined that we would present participants with 100 items, ten of which were included to test the hypothesis discussed in Sloman et al. (under review) and two of which were included as part of our attention check.

4. Fourth, please add a sentence within the manuscript that provides a reference to the location of your data and code (and please make sure that the studies in this manuscript are aligned with what is contained on your GitHub page).

We include the link to the GitHub repository on line 277. We have also updated the repository to align with the analyses and results reported in the submitted version of the manuscript. 

We hope our revisions adequately address your concerns. We thank you and the reviewer for the time taken to provide feedback on our work.

Sincerely,

Sabina J. Sloman

Dr. Daniel Oppenheimer

Dr. Simon DeDeo

References

Sloman, S.J., Oppenheimer, D., DeDeo, S. (under review). One Fee, Two Fees; Red Fee, Blue Fee: People Use the Valence of Others’ Speech in Social Relational Judgments.

---

## [Editor Report · Decision Letter 2]

25 Jan 2021

Can we detect conditioned variation in political speech? Two kinds of discussion and types of conversation

PONE-D-20-30828R2

Dear Dr. Sloman,

We’re pleased to inform you that your manuscript has been judged scientifically suitable for publication and will be formally accepted for publication once it meets all outstanding technical requirements.

Kind regards,

Thomas Holtgraves, Ph.D.

Academic Editor

PLOS ONE
---

## [Editor Report · Acceptance letter]

1 Feb 2021

PONE-D-20-30828R2 

Can we detect conditioned variation in political speech? Two kinds of discussion and types of conversation 

Dear Dr. Sloman:

I'm pleased to inform you that your manuscript has been deemed suitable for publication in PLOS ONE. Congratulations! Your manuscript is now with our production department. 

Kind regards, 

on behalf of

Dr. Thomas Holtgraves 

Academic Editor

PLOS ONE